# StructEval: A Benchmark for Evaluating Generation, Inference, and Reconstruction in Atomic and Crystalline Structures

## Abstract

Crystalline materials power technologies from solar conversion to catalysis, yet current machine learning evaluations artificially divide infinite lattices and finite nanoclusters into separate domains. StructEval unifies these regimes with a symmetry-aware, radius-resolved framework that systematically links primitive unit cells to nanoparticles across ten industrially critical compounds. Each material includes 20+ radii configurations between 0.6–3.0 nm, sampled at 780 quasi-uniform orientations, producing nearly $200,000$ structures spanning $55 - 11,298$ atoms. StructEval defines three rigorous challenges–unit cell $\rightarrow$ nanoparticle generation, nanoparticle $\rightarrow$ unit cell resolution, and lattice reconstruction–supported by a leakage-free split that isolates orientation interpolation from radius extrapolation. This design enables precise measurement of generalization across scale and symmetry. Benchmarking leading generative models reveals severe breakdowns under out-of-distribution conditions, exposing a fundamental gap in current architectures. By providing a reproducible, geometry-grounded testbed, StructEval establishes the foundation for next-generation generative, inference, and reconstruction models in crystalline systems. Data and implementations are released at `https://anonymous.4open.science/r/StructEval-ANONYMOUS`.

## 1 Introduction

Nanostructured materials enable applications from photovoltaics to sensing Simonov & Goodwin (2020), with behavior governed both by the periodicity of a primitive unit cell and by the finite morphologies that emerge in nanoparticles Cao et al. (2022). These regimes are typically studied separately: crystal-growth simulations extend ideal lattices, whereas nanoparticle workflows build finite clusters refined by empirical or ab-initio methods Levi & Kotrla (1997). However, such simulations are computationally expensive and difficult to scale across compositions, sizes, and orientations Surek (2005). First-principles approaches such as DFT Orio et al. (2009) and DFTB Elstner & Seifert (2014) provide accurate energetics and reconstructions, but DFT's cubic cost limits large-scale exploration Cohen et al. (2008); DFTB partially alleviates this Liu et al. (2019); Qi et al. (2013).

Machine learning (ML) offers a scalable alternative Carleo et al. (2019); Karniadakis et al. (2021); Alpaydin (2021). GNNs including SchNet Schütt et al. (2018), DimeNet++ Gasteiger et al. (2020), TorchMD-Net Thölke & De Fabritiis (2022), and CGNNs Cheng et al. (2021) achieve strong molecular and crystalline prediction performance on benchmarks such as QM9 Ruddigkeit et al. (2012); Ramakrishnan et al. (2014) and MD17 Chmiela et al. (2017). Equivariant models—E(n)-GNNs Satorras et al. (2021), SE(3)-Transformers Fuchs et al. (2020), PaiNN Schütt et al. (2021), SphereNet Coors et al. (2018), Equiformer Liao & Smidt (2022), FAENet Duval et al. (2023), and GotenNet Aykent & Xia (2025)—improve data efficiency by enforcing geometric symmetries, while multimodal approaches Polat et al. (2024); Rollins et al. (2024); Das et al. (2023) integrate diverse inputs. Generative models such as CDVAE Xie et al. (2021), GraphDF Luo et al. (2021), UniMat Yang et al. (2023), and CrystalFlow Luo et al. (2024), along with recent symmetry-preserving diffusion and flow models Jiao et al. (2024); Levy et al. (2025); Kelvinius et al. (2025), advance crystal generation but are primarily evaluated on small molecules or bulk periodic crystals. No existing benchmark links

primitive-cell inputs to radius- and orientation-controlled nanoparticles or tests generalization under realistic shifts in scale and symmetry.

StructEval addresses this gap with a symmetry-aware, radius-resolved dataset connecting primitive unit cells to systematically generated nanoparticles. For ten technologically relevant materials (Ag, Au, $CH_3NH_3PbI_3$, $Fe_2O_3$, $MoS_2$, PbS, $SnO_2$, $SrTiO_3$, $TiO_2$, ZnO), the dataset provides each unit cell, 25 radii from 0.6–3.0 nm, and 780 quasi-uniform orientations, yielding $\sim$200,000 reproducible structures containing 55–11,298 atoms. StructEval defines three tasks: (1) unit-cell $\rightarrow$ nanoparticle generation, (2) nanoparticle $\rightarrow$ lattice inference, and (3) partial-structure reconstruction. A leakage-free split ensures interpolation over orientations (ID) and extrapolation across radii (OOD) are cleanly separated. Evaluations of modern generative models—including DiffCSP Jiao et al. (2023), FlowMM Miller et al. (2024), FlowLLM Sriram et al. (2024), MatterGen-MP Zeni et al. (2023), and ADiT Joshi et al. (2025)—show consistent degradation under OOD shifts, highlighting challenges in symmetry preservation, scale generalization, and coherent reconstruction. By unifying bulk crystallography with nanoparticle geometry, StructEval provides a rigorous, reproducible platform for benchmarking generative and inference models under realistic structural distribution shifts.

The remainder of the paper is organized as follows. Section 2 presents a review of prior work on crystalline building blocks and nanostructured morphologies, including first-principles and semi-empirical simulation methods, as well as existing benchmarks for crystals and nanomaterials. Section 3 details the construction and characteristics of the StructEval dataset. Section 4 discusses the performance of the SOTA generative models on StructEval across three distinct tasks. Section 5 outlines the current limitations of StructEval dataset. Section 6 summarizes the key findings and proposes directions for future research.

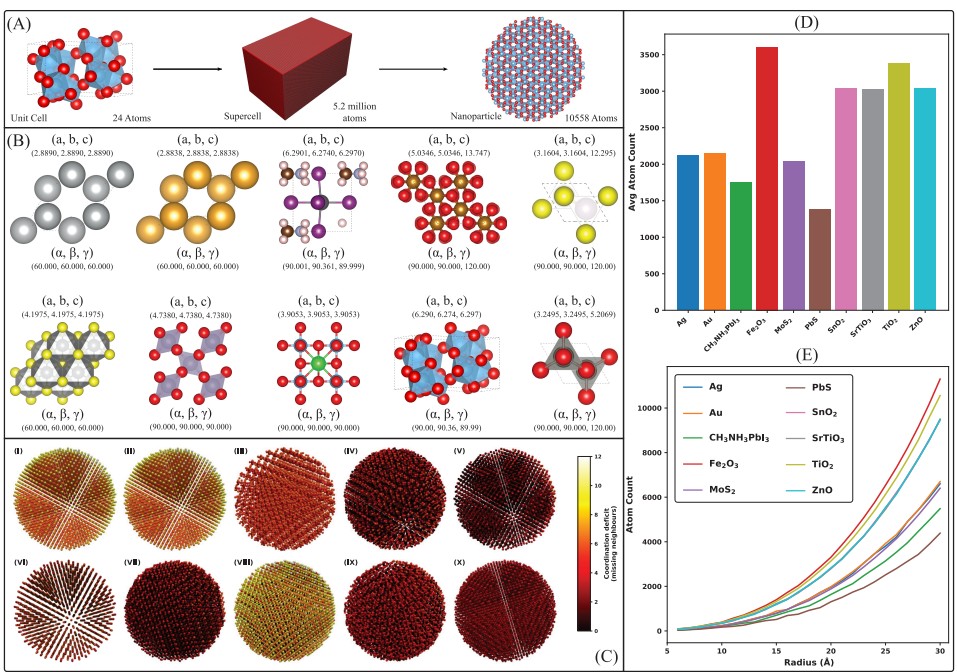

Figure 1: Detailed overview of StructEval. (A) Workflow for generating radius-resolved nanoparticles. (B) Unit cells of the materials—Ag, Au, $CH_3NH_3PbI_3$, $Fe_2O_3$, $MoS_2$, PbS, $SnO_2$, $SrTiO_3$, $TiO_2$, and ZnO—arranged left to right. The lattice constants a, b and c denote the cell edge lengths, while the angles $\alpha$, $\beta$ and $\gamma$ specify the inter-edge angles between b–c, a–c and a–b, respectively. (C) Coordination-deficit heatmaps for the largest nanoparticle of each material (panels I–X follow the same material order). (D) Mean atom count per material, illustrating typical nanoparticle sizes. (E) Atom count versus nanoparticle radius, revealing how total atom number scales with size.

## 2 RELATED WORK

### 2.1 CRYSTALLINE BUILDING BLOCKS AND NANOSTRUCTURED MORPHOLOGIES

Crystalline unit cells encode the symmetry operations and atomic motifs that ML models must learn to generalize, providing the periodic, coordinative, and orientational patterns from which larger structures derive. When bulk crystals are truncated into finite clusters, their shapes follow orientation-dependent surface energies described by the Gibbs–Wulff theorem Li et al. (2016); Barmparis et al. (2015); Ringe et al. (2013). In StructEval, these effects matter not for their thermodynamic meaning but because changes in radius, orientation, and faceting produce controlled, symmetry-consistent deviations from ideal bulk geometry Kittel & McEuen (2018), forming principled distribution shifts for evaluating geometric generalization. Quantum-confinement–related optical or catalytic phenomena Bera et al. (2010) lie outside the benchmark's scope; only the crystallographic information needed to produce meaningful radius-resolved nanoclusters is retained. By using deterministic, symmetry-preserving construction rather than simulation-driven morphologies, StructEval offers reproducible and physically grounded structural variation tailored for ML benchmarking Yang et al. (2022).

### 2.2 FIRST-PRINCIPLES AND SEMI-EMPIRICAL SIMULATION METHODS

Physics-based nanoparticle generation methods—including Kohn–Sham DFT Bickelhaupt & Baerends (2000); Yu et al. (2016), linear-scaling approaches such as ONETEP Baer & Head-Gordon (1997); Skylaris et al. (2005), semi-empirical techniques like DFTB Zheng et al. (2005); Spiegelman et al. (2020); Bačić et al. (2020); Kim et al. (2019), and classical or ML-based interatomic potentials Daw & Baskes (1984); Behler & Parrinello (2007); Mahata et al. (2022)—accurately capture energetics and surface reconstructions but face steep size limitations. Standard DFT's $\mathcal{O}(N^3)$ scaling restricts simulations to $\sim 10^3$ atoms, and even reduced-scaling or semi-empirical models cannot reliably generate structures near StructEval's 11,300-atom upper bound. Because StructEval focuses on geometric and symmetry-based reasoning rather than energetics, fully physics-based generation is unnecessary and computationally infeasible. Instead, deterministic symmetry-preserving construction provides scalable, reproducible structures that maintain crystallographic motifs while enabling systematic variation of radius and morphology. This rationale explains why StructEval employs geometric rather than DFT-, DFTB-, or force-field–driven workflows and highlights that geometric generalization—not simulation fidelity—is the benchmark's core objective Kurban et al. (2024).

### 2.3 BENCHMARKS FOR CRYSTALS AND NANOMATERIALS

Benchmark datasets have propelled ML advances in quantum chemistry and materials science by supplying standardized structures and labels. Beyond those noted in Section 1, QM7 Blum & Reymond (2009); Rupp et al. (2012) provides atomization energies for small organic molecules, MD22 Chmiela et al. (2023) offers force-labeled molecular dynamics trajectories, and PubChemQC Kim et al. (2025) furnishes millions of ground-state geometries with electronic properties. NablaDFT Khrabrov et al. (2022) and QH9 Yu et al. (2024) target Hamiltonian matrix estimation, while Perov-5 Castelli et al. (2012a;b) includes 18,928 $ABX_3$ perovskites and Carbon-24 Pickard (2020) catalogs over 10,000 low-energy carbon frameworks generated by AIRSS and relaxed via DFT. For inorganic solids, the Materials Project underpins MatBench's fourteen property benchmarks Dunn et al. (2020), and the OC20 Chanussot et al. (2021) and OC22 Tran et al. (2023) datasets provide force-labeled metal and oxide surface structures. LAMBench Peng et al. (2025) aggregates diverse atomic geometries for large-scale pretraining, while CrysMTM Polat et al. (2025) introduces temperature-dependent nanomaterial configurations with multimodal annotations. Despite this breadth, existing datasets focus mainly on bulk crystals or small molecules; radius-graded nanoclusters typically appear only incidentally in catalytic or surface studies, and symmetry-aware or bidirectional tasks remain rare. No public benchmark systematically couples primitive-cell representations to radius- and orientation-resolved nanoclusters across chemistries and size regimes, nor supports tasks such as nanoparticle generation or lattice inference from finite clusters—motivating the development of StructEval.

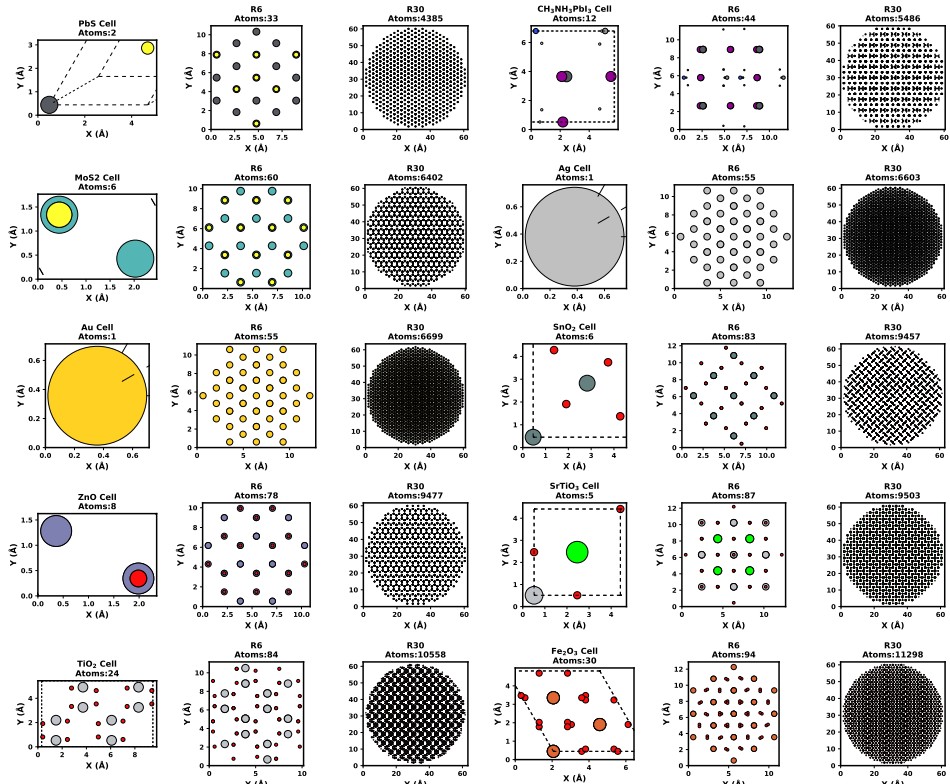

Figure 2: From primitive cell to radius-controlled nanoclusters. For each material in dataset, the panels show—left to right—the primitive unit cell followed by its canonical R = 6 Å and R = 30 Å nanoparticles. Materials are arranged from top to bottom in ascending order of the atom count in their R30 cluster, illustrating how coordination environments and bulk-like cores emerge with increasing radius. All views share a common Ångström scale. Atom colours follow the conventional CPK palette: Ag (light grey), Au (gold), C (dark grey), H (white), N (blue), O (red), S (yellow), Fe (orange-brown), Mo (teal), Sn (silver), Sr (green), Ti (slate grey), Zn (forest green), Pb (dark grey-blue), and I (purple).

## 3 DATASET CONSTRUCTION AND TASK DEFINITION

### 3.1 CRYSTAL STRUCTURES AND NANOPARTICLE GENERATION

The STRUCTEVAL suite spans elemental solids, perovskites, transition-metal dichalcogenides, and binary oxides. The selected compounds include Silver (Ag) King (2002b), Gold (Au) King (2002a), methylammonium lead iodide ($CH_3NH_3PbI_3$) Walsh et al. (2019), hematite ($Fe_2O_3$) Finger & Hazen (1980), molybdenum disulfide ($MoS_2$) Wyckoff (1963a); Grau-Crespo & Lopez-Cordero (2002), galena (PbS) Wyckoff (1963b), cassiterite ($SnO_2$) Baur et al. (1971), strontium titanate ($SrTiO_3$) Mitchell & Carpenter (2000), anatase titanium dioxide ($TiO_2$) Horn et al. (1972), and zincite zinc oxide (ZnO) Wyckoff (1963c). Primitive unit cells were extracted from CIFs and serve as the basis for nanoparticle generation and symmetry-aware modeling tasks. Lattice parameters, space groups, and atomic positions are provided in Appendix A, while Figure 2 summarizes dataset structure and key statistics.

**Supercell construction.** Let $N = 60$. We define the periodic lattice of atomic sites as

$$\Lambda = \left\{ n_1\mathbf{a}_1 + n_2\mathbf{a}_2 + n_3\mathbf{a}_3 + \mathbf{r}_j \;\middle|\; n_1, n_2, n_3 \in \{0, \dots, N-1\}, \; j \in \{1, \dots, M\} \right\}, \quad (1)$$

where $\mathbf{a}_1, \mathbf{a}_2, \mathbf{a}_3 \in \mathbb{R}^3$ are the primitive lattice vectors of the unit cell, and $\{\mathbf{r}_j\}_{j=1}^M \subset \mathbb{R}^3$ are the basis positions of the $M$ atoms inside each unit cell. The integers $n_1, n_2, n_3 \in \{0, 1, \dots, N-1\}$

serve as lattice indices, specifying the unit cell position along the three lattice directions, and may equivalently be viewed as elements of $\mathbb{Z}_N$. Replicating the unit cell $N$ times along each direction yields an $N \times N \times N$ supercell, and the set $\Lambda$ in equation 1 collects all atomic sites contained within it.

**Reference atom.**   Choose a central reference position $\mathbf{r}_0 \in \Lambda$ (e.g., the image of $\mathbf{r}_1$ in the central unit cell defined by equation 1).

**Nanoparticle definition.**   For a given cutoff radius $R$, define the nanoparticle as

$$P(R) \;=\; \Big\{ \mathbf{x} \in \Lambda \;\Big|\; \|\mathbf{x} - \mathbf{r}_0\|_2 \leq R \Big\}, \tag{2}$$

i.e., all lattice sites within a sphere of radius $R$ around $\mathbf{r}_0$.

**Radius sampling.**   Select $K = 25$ radii on a uniform grid:

$$R_k \;=\; R_{\min} + (k - 1)\,\Delta R, \qquad k = 1, \dots, K, \tag{3}$$

with end-points and step size

$$R_{\min} = 0.6\,\text{nm}, \qquad R_{\max} = 3.0\,\text{nm}, \qquad \Delta R \;=\; \frac{R_{\max} - R_{\min}}{K - 1} \;=\; 0.1\,\text{nm}. \tag{4}$$

Relationships captured by the equations 2–4 define the complete set $\{P(R_k)\}_{k=1}^{25}$, spanning quantum-confined clusters ($R \leq 1.0\,\text{nm}$) to near-bulk fragments ($R \geq 2.0\,\text{nm}$).

### 3.2 Quasi-Uniform Rotations

Deterministic, quasi-uniform rotations ensure that each nanoparticle orientation is represented without randomness, enabling reproducible data augmentation and unbiased evaluation of rotation-equivariant models. By constructing a finite, low-discrepancy *set* on SO(3) that includes the identity and approximates uniform coverage, one can systematically generate rotated copies of any atomic configuration Yershova et al. (2010). Let $\mathcal{P} = \{\mathbf{r}_1, \dots, \mathbf{r}_n\} \subset \mathbb{R}^3$ denote a finite point cloud (with $n$ points). Our goal is to build a finite set $\mathcal{G} \subset$ SO(3) containing the identity that provides near-uniform, reproducible sampling of rotation space. The construction proceeds in three steps:

**Axis–angle parametrization.**   Every rotation $R \in$ SO(3) is represented as

$$R \;=\; R(\hat{\mathbf{u}}, \phi), \tag{5}$$

where $\hat{\mathbf{u}} \in \mathbb{S}^2$ is a unit axis and $\phi \in [0, \pi]$ is the rotation angle, with the equivalence $(\hat{\mathbf{u}}, \phi) \sim (-\hat{\mathbf{u}}, 2\pi - \phi)$ ensuring a unique representation. Using Rodrigues' formula Dai (2015), this rotation is expressed as

$$R(\hat{\mathbf{u}}, \phi) = I_3 \cos \phi + (1 - \cos \phi)\,\hat{\mathbf{u}}\,\hat{\mathbf{u}}^\top + \sin \phi\,[\hat{\mathbf{u}}]_\times, \tag{6}$$

where $I_3$ is the $3 \times 3$ identity matrix, $\hat{\mathbf{u}} = (u_1, u_2, u_3)^\top$, and

$$[\hat{\mathbf{u}}]_\times \;=\; \begin{pmatrix} 0 & -u_3 & u_2 \\ u_3 & 0 & -u_1 \\ -u_2 & u_1 & 0 \end{pmatrix} \tag{7}$$

is the skew-symmetric matrix implementing the linear map $\mathbf{v} \mapsto \hat{\mathbf{u}} \times \mathbf{v}$, which guarantees that $R(\hat{\mathbf{u}}, \phi)$ in equation 6 is orthogonal and has determinant 1.

**Quasi-Uniform Grid on** SO(3)**.**   We define a deterministic sampling scheme via the Hopf fibration Lyons (2003):

$$(\hat{\mathbf{u}}, \psi) \;\longmapsto\; R(\hat{\mathbf{u}}, 2\psi), \qquad \hat{\mathbf{u}} \in \mathbb{S}^2, \ \psi \in [0, \pi).$$

To discretize this space, we proceed as follows.

    1. **Fibonacci lattice on** $\mathbb{S}^2$**.** For $k = 0, \dots, N_{\text{axis}} - 1$ set

$$z_k = 1 - \frac{2k + 1}{N_{\text{axis}}}, \quad r_k = \sqrt{1 - z_k^2}, \quad \varphi_g = \frac{1 + \sqrt{5}}{2}, \quad \varphi_k = 2\pi k\,\varphi_g^{-1},$$

and define the axis $\hat{\mathbf{u}}_k = (r_k \cos \varphi_k, \ r_k \sin \varphi_k, \ z_k)$.

2. **Irrational angle increments.** For $m = 0, \ldots, M_{\text{ang}} - 1$, let

$$\psi_m = 2\pi m \, \varphi_g^{-1} \mod 2\pi.$$

3. **Rotation-set construction.** Define $R_{k,m} = R\big(\hat{\mathbf{u}}_k, \, 2\psi_m\big)$ and let $\mathcal{G}$ be the set consisting of the identity and all $R_{k,m}$. Because $\psi_0 = 0$ yields $R = I$, we count identity once, hence

$$|\mathcal{G}| = 1 + N_{\text{axis}} (M_{\text{ang}} - 1).$$

**Quasi-Uniform Coverage (sketch).**  As $N_{\text{axis}}, M_{\text{ang}} \to \infty$, the induced empirical measure of the above low-discrepancy set provides increasingly uniform coverage of $\text{SO}(3)$ with respect to the Haar measure (in the sense of vanishing discrepancy), leveraging the equidistribution of the Fibonacci lattice on $\mathbb{S}^2$ Stanley (1975) and the incommensurate twist increments John (1998).

**Rigid-body action on the point cloud.**  Let $\mathcal{P} = \{\mathbf{r}_1, \ldots, \mathbf{r}_n\} \subset \mathbb{R}^3$ be a finite point cloud and $\mathbf{c} = \frac{1}{n} \sum_{i=1}^{n} \mathbf{r}_i$ be its center of mass. For each $R \in \mathcal{G}$, define the rigidly rotated configuration

$$\mathcal{P}_R = \big\{ R(\mathbf{r}_i - \mathbf{c}) + \mathbf{c} \ \big| \ i = 1, \ldots, n \big\}, \tag{8}$$

which preserves all inter-point distances while re-centering about $\mathbf{c}$. To ensure numerical robustness, a tolerance $\varepsilon > 0$ is used when applying equation 8: near-zero angles are snapped, degenerate axes are regularized, and infinitesimal perturbations break floating-point degeneracies. This deterministic augmentation yields a near-uniform sampling of orientations from $\mathcal{G}$ without introducing stochastic noise.

**Configuration diversity.**  Although individual nanoparticles may exhibit high symmetry, the Cartesian product of (i) ten distinct chemistries/crystal families, (ii) 25 radii spanning quantum-confined to near-bulk regimes, and (iii) $|\mathcal{G}| = 780$ deterministic orientations produces $\approx 200{,}000$ unique atomic configurations, capturing rich variation in surface terminations, coordination statistics, facet exposure, and global morphology—especially for non-cubic/anisotropic lattices.

### 3.3 Data Splits

For each material $m$, we enumerate all nanoparticle instances as

$$\mathcal{G}_m = \{(R, b) \mid R \in \{\text{R6}, \ldots, \text{R30}\}, \ b \in \{0, \ldots, 780\}\},$$

covering every radius–orientation pair. We treat radii R6–R24 as in-distribution and R25–R30 as out-of-distribution, then apply a random split (70%/10%/20%) to the in-distribution subset to obtain $\text{Train}_m$, $\text{Val}_m$, and $\text{IDTest}_m$, with $\text{OODTest}_m$ defined by the held-out radii. Grouping all orientations $(R, b)$ together ensures rotation-invariant partitions and prevents orientation leakage.

**Physical grounding and scope**. Nanoparticles are generated by deterministic spherical truncation of experimentally characterized bulk crystals, preserving local crystallography while breaking global periodicity. We omit thermodynamic or electronic labels and avoid large-scale ab-initio relaxations to keep the benchmark geometry- and symmetry-focused, fully reproducible at $\approx 200{,}000$ structures. Because DFT scales cubically and our clusters reach $\approx 10^3$–$10^4$ atoms, exhaustive first-principles labeling is infeasible; instead, we rely on a symmetry-aware geometric pipeline that cleanly supports ID/OOD splits. **On rotations**. Rotations are central to defining orientation generalization: for each radius $R_k$, we construct non-overlapping train/ID/OOD $\text{SO}(3)$ grids with decreasing angular spacing $(9°, 6°, 3°)$ using a greedy quaternion scheme and store rotated copies indexed by their quaternion. We do not apply Kabsch or ICP alignment, since alignment removes the requirement to predict absolute orientation and would inflate scores for models that produce consistently rotated—but not correctly oriented—structures.

### 3.4 Tasks

The dataset spans three tasks evaluated with metrics that capture both geometric and crystallographic fidelity. Core geometric measures include RMSD, Hausdorff distance Rucklidge (1997), convex-hull volume ratio error, radial-distribution-function divergence, and validity rate. Task 2 additionally evaluates lattice-parameter error, space-group accuracy, and their joint accuracy. Task 3 further

includes surface-atom recall and coordination-histogram divergence. All tasks rely on the material-specific splits $\text{Train}_m$, $\text{Val}_m$, $\text{IDTest}_m$, and $\text{OODTest}_m$ to separately assess orientation interpolation and radius extrapolation. Figure 3 summarizes the three tasks, with detailed metric definitions provided in Appendix B.

**Task 1: Unit Cell $\rightarrow$ Nanoparticle Generation.** From a primitive unit cell $u_m$ and a target radius $R$, the model must generate realistic nanoparticles that faithfully reproduce both the infinite lattice periodicity and the finite-size surface morphology. Mastering this capability is essential for producing computationally efficient nanostructure models that accurately capture surface-driven phenomena, thereby enhancing predictive accuracy in material design:

$$f_1 : (u_m, R) \longmapsto P \subset \mathbb{R}^3. \tag{9}$$

The mapping in equation 9 is assessed with the metrics $\text{RMSD}(P, P^*)$, $d_{\text{Haus}}(P, P^*)$, $\Delta V_{\text{hull}}(P, P^*)$, $E_{\text{RDF}}(P, P^*)$, and $\text{V}_{\text{R}}(P)$.

**Task 2: Nanoparticle $\rightarrow$ Lattice Inference.** Local geometric information extracted from finite nanoparticles is used to infer their lattice constants and symmetry group, exploring whether such short-range cues suffice to reconstruct extended periodicity and crystallographic invariants. Achieving this capability is vital for bridging localized observations with global structural order, thereby enabling accurate property predictions and reliable materials-design workflows:

$$f_2 : P \longmapsto (\ell, g), \quad \ell = (a, b, c, \alpha, \beta, \gamma), \quad g \in \mathcal{S}, \tag{10}$$

so that equation 10 maps each nanoparticle $P$ to its lattice parameters $\ell$ and a space-group label $g$ drawn from the set $\mathcal{S}$ of crystallographic groups. Evaluation metrics are then $\text{RMSE}(\ell, \ell^*)$, $\mathbb{1}[g = g^*]$, and the joint accuracy $\mathbb{1}[(\ell, g) = (\ell^*, g^*)]$, all assessed with respect to the ground-truth pair $(\ell^*, g^*)$.

**Task 3: Lattice Reconstruction.** Partial occlusion of nanoparticles—whether applied at random or biased toward surface atoms—tests the model's capability to reconstruct the complete atomic arrangement from incomplete inputs. Such evaluation reveals whether the model can infer missing structural information and preserve chemical fidelity, a critical requirement for reliable nanoparticle modeling and downstream materials-design applications:

$$f_3 : P_{\text{mask}} \mapsto P, \quad P_{\text{mask}} = M \cdot P, \tag{11}$$

where $M$ in Eq. (11) is a masking operator that removes a fraction $p$ of atoms. Evaluation metrics include $\text{RMSD}(P, P^*)$, $\text{R}_{\text{surf}}(P, P^*)$, $D_{\text{KL}}(h(P) \,\|\, h(P^*))$, and $\text{V}_{\text{R}}(P)$.

## 4 EXPERIMENTS

Multiple generative models—such as DiffCSP, FlowMM, FlowLLM, MatterGen-MP, and ADiT—are evaluated on all three tasks using the splits defined in Section 3.4. Implementation details and hyperparameters are provided in Appendix C, with comprehensive task-specific results in Appendix D. To clarify task solvability, we note that Task 1 (unit cell $\rightarrow$ nanoparticle) is deterministic under spherical truncation of the periodic lattice; Task 2 (nanoparticle $\rightarrow$ lattice inference) remains solvable because local neighborhoods preserve crystallographic invariants such as coordination shells, bond-angle distributions, and stoichiometry; and Task 3 (partial reconstruction) has a clear lower bound, as oracle-style baselines that fill masked regions via nearest-neighbor lattice completion achieve substantially lower errors than learned models, indicating significant remaining headroom.

### 4.1 TASK 1: UNIT CELL TO NANOPARTICLE GENERATION

Task 1 evaluates nanoparticle generation from a primitive unit cell and target radius equation 9, requiring models to capture both lattice periodicity and finite-size morphology. As shown in Table 1, no method excels across normalized RMSD, RDF KL divergence, and volume ratio in either ID or OOD settings: MatterGen-MP attains the lowest RMSD but suffers from large volume ratios; FlowLLM achieves the best $\text{RDF}_{\text{KL}}$ but with high RMSD; and DiffCSP and FlowMM provide mid-range RMSD while lagging on other metrics. OOD performance is evaluated using leakage-free splits

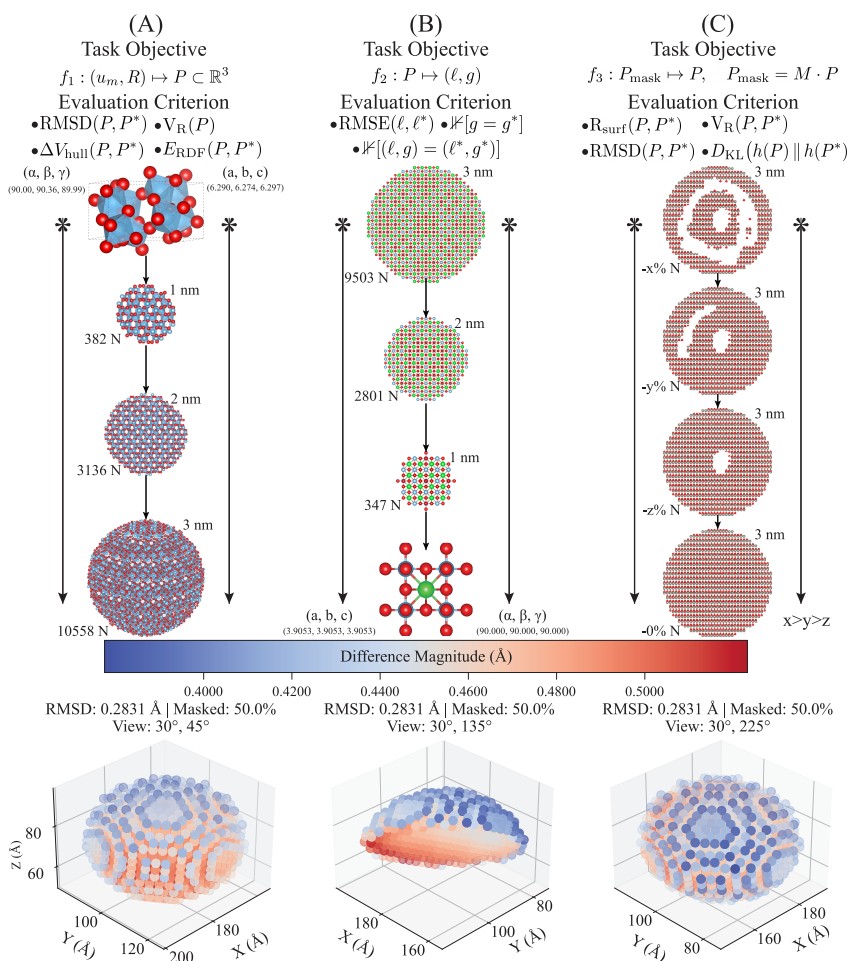

Figure 3: Overview of benchmark tasks and lattice reconstruction experiment result. (A) shows generation of nanoparticles starting from the unit cell. (B) includes the generating unit cell configurations from the nanoparticles. Lastly, (C) illustrates lattice reconstruction from partially masked configurations. N is the atom count in the system. First material is $TiO_2$, second is $SrTiO_3$ while the last one is ZnO. Scatter plots in the bottom are different angle reconstruction results from CDVAE model for a single material on OOD set with masked 50% of the original atoms.

that exclude entire radius ranges and deploy disjoint orientation grids, ensuring models are tested on true scale and orientation extrapolation. The consistent OOD degradation across architectures indicates limitations in scale awareness, multi-scale feature modeling, and long-range symmetry handling. Training times further reveal large efficiency gaps—FlowLLM is orders of magnitude slower per epoch, while faster models such as MatterGen-MP and DiffCSP do not achieve the best overall accuracy. These results underscore fundamental trade-offs between geometric fidelity, physical realism, and computational efficiency, and point to the need for explicit radius conditioning, hierarchical representations, and stronger geometric inductive biases.

## 4.2 Task 2: Nanoparticle to Lattice Inference

Task 2 evaluates lattice-parameter and space-group inference from finite nanoparticles equation 10, testing whether local geometry is sufficient to recover global crystallographic invariants. As shown in Table 2, models such as DiffCSP and ADiT achieve relatively low lattice-parameter RMSEs but perform poorly on symmetry classification, with all space-group accuracies below 40% and joint accuracy effectively 0%. For example, ADiT attains an ID RMSE of 55.240 Å, 29.6% space-group accuracy, and 0.0% joint accuracy; OOD results remain similar at 55.246 Å, 29.1%, and 0.0%.

Because each material's space group is constant across radii, these accuracies reflect per-material label learning, while the stable RMSE shows that lattice-parameter prediction generalizes to larger clusters. The persistent $0\%$ joint accuracy highlights the core difficulty: nanoparticles preserve only local symmetry, losing global periodicity under spherical truncation, making simultaneous recovery of lattice parameters and exact space group inherently ambiguous. OOD generalization remains limited, and training-time disparities further emphasize model constraints—FlowLLM is computationally prohibitive without accuracy gains, whereas faster models like DiffCSP and ADiT still fail to infer symmetry reliably. Overall, current architectures partly recover unit-cell geometry but cannot translate local structure into correct global symmetry, which is precisely the challenge Task 2 is designed to expose.

## 4.3 TASK 3: LATTICE RECONSTRUCTION

Task 3 evaluates a model's ability to reconstruct full nanoparticle structures from partially observed inputs equation 11, using random and surface-biased masking to test whether missing atoms can be recovered while maintaining structural and chemical fidelity. As shown in Table 3, performance is uniformly poor: RMSD remains high, surface-atom recall is near zero, and validity rates rarely exceed $0\%$. Even the strongest models—FlowMM and FlowLLM—offer only marginal improvements, failing to restore missing surface atoms or preserve local chemistry; RDF KL divergence also stays large across all methods, indicating that none recover the underlying crystalline order from incomplete data. MatterGen-MP and ADiT show slightly lower divergence in isolated cases, but no model demonstrates consistent or robust reconstruction. Training-time comparisons further reveal substantial inefficiencies: ADiT is faster but not more accurate, while FlowLLM is both slow and ineffective. Overall, greater model capacity or compute does not overcome the inherent difficulty of nanoparticle completion, and the persistent failures highlight why Task 3 serves as a stringent benchmark for partial-structure recovery; an illustrative reconstruction error map is shown in Figure 3.

## 5 LIMITATIONS

While StructEval bridges bulk symmetry and finite nanoclusters, it has limitations: it covers only ten materials currently; our clusters are crystallographically grounded fragments—deterministic truncations of experimentally validated bulk structures—but they are not thermodynamically validated or passivated, and they omit temperature, disorder, and environmental effects. This is intentional: by decoupling geometric/symmetry reasoning from energetic confounders, the benchmark provides a clean, deterministic testbed for generation, inference, and reconstruction under strict ID/OOD controls. We acknowledge that experimental nanoparticles can undergo surface reconstructions and environment-dependent stabilization.

## 6 CONCLUSION AND FUTURE WORK

Existing benchmarks emphasize either periodic crystals or small molecules, leaving the transitional bulk-to-nanoscale regime largely unaddressed. StructEval fills this gap with a radius-resolved, symmetry-aware dataset linking primitive unit cells to nanoparticles sampled across 25 radii and 780 quasi-uniform orientations, supporting three bidirectional tasks—generation, inference, and reconstruction—under strict leakage-free splits that probe both interpolation and extrapolation. Although the benchmark focuses on real-space geometric and symmetry reasoning, it remains fully compatible with XRD-style structure-solution tasks Guo et al. (2025), as each nanoparticle configuration can be used to simulate nanocrystalline diffraction patterns. Across tasks, state-of-the-art models perform reasonably in-distribution but degrade sharply out-of-distribution, exhibiting increased reconstruction errors, symmetry misclassifications, and completion failures. Since DFT scales cubically with system size, direct simulation of large nanoparticles is impractical, and surrogate models that fail to generalize cannot reliably replace it for predicting lattice parameters or morphologies. These findings highlight the limitations of relying solely on local geometric cues and motivate approaches with explicit symmetry constraints, multi-scale hierarchies, and targeted augmentation. Designed as a modular platform with planned extensions to molecular crystals, perovskites, polymers, and multi-phase systems, StructEval unifies bulk and nanoscale regimes and establishes a foundation for developing models capable of capturing the full structural complexity of crystalline nanomaterials.

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
