## A   DFTB SIMULATION DETAILS

Here are the DFTB simulation parameters for each material used in the dataset.

**Silver (Ag).**   Silver crystallizes in a face-centered cubic (FCC) structure with lattice constant $a = 4.0857$ Å. The crystal system is cubic with space group Fm$\bar{3}$m (No. 225), Pearson symbol cF4, and Schoenflies notation $O_h^5$. A single Ag atom is located at the origin of the primitive cell King (2002b).

**Gold (Au).**   Gold also forms an FCC structure with a slightly smaller lattice constant $a = 4.0782$ Å. It belongs to the same cubic space group Fm$\bar{3}$m (No. 225), reflecting high symmetry. One gold atom is positioned at (0, 0, 0) within the unit cell King (2002a).

**Methylammonium Lead Iodide (CH$_3$NH$_3$PbI$_3$).**   This hybrid perovskite exhibits a pseudo-cubic lattice with parameters $a = 6.290$ Å, $b = 6.274$ Å, $c = 6.297$ Å and angles close to $90°$. The space group is $P1$ (No. 1), capturing slight distortions and dynamic disorder typical in organic-inorganic perovskites Walsh et al. (2019).

**Hematite (Fe$_2$O$_3$).**   Fe$_2$O$_3$, known as hematite, adopts a rhombohedral structure with lattice constants $a = b = 5.0346$ Å, $c = 13.7473$ Å and angles $\alpha = \beta = 90°, \gamma = 120°$. It crystallizes in the $R\bar{3}c$ space group (No. 167), which enables antiferromagnetic and catalytic properties Finger & Hazen (1980).

**Molybdenum Disulfide (MoS$_2$).**   MoS$_2$ is a layered transition metal dichalcogenide with hexagonal lattice parameters $a = 3.1604$ Å, $c = 12.295$ Å, and angles $\alpha = \beta = 90°, \gamma = 120°$. It crystallizes in the space group $P6_3/mmc$ (No. 194), reflecting its layered van der Waals structure Wyckoff (1963a); Grau-Crespo & Lopez-Cordero (2002).

**Galena (PbS).**   Lead sulfide adopts a rock-salt type FCC structure with lattice constant $a = 5.9362$ Å. The space group is Fm$\bar{3}$m (No. 225), and the crystal system is cubic. Pb and S atoms occupy alternating FCC lattice points Wyckoff (1963b).

**Cassiterite (SnO$_2$).**   SnO$_2$ has a tetragonal rutile-type structure with lattice constants $a = 4.738$ Å, $c = 3.1865$ Å. It crystallizes in the space group $P4_2/mnm$ (No. 136), and features a complex oxygen sublattice surrounding tin atoms Baur et al. (1971).

**Strontium Titanate (SrTiO$_3$).**   SrTiO$_3$ forms a cubic perovskite structure with lattice parameter $a = 3.9053$ Å, and belongs to space group $Pm\bar{3}m$ (No. 221). The ideal symmetry makes it a model material for ferroelectric and quantum paraelectric studies Mitchell & Carpenter (2000).

**Titanium Dioxide (TiO$_2$ - Anatase phase).**   Anatase TiO$_2$ crystallizes in a body-centered tetragonal structure with $a = 3.7842$ Å and $c = 9.5146$ Å. The space group is $I4_1/amd$ (No. 141), typical of the anatase polymorph with photocatalytic functionality Horn et al. (1972).

**Zinc Oxide (ZnO - Zincite phase).**   ZnO exhibits a hexagonal wurtzite-type structure with lattice constants $a = 3.2495$ Å, $c = 5.2069$ Å, and space group $P6_3mc$ (No. 186). This polar crystal system plays a key role in piezoelectric and optoelectronic devices Wyckoff (1963c).

The dataset generation process can be described at a more abstract level as follows:

1. **Unit Cell Acquisition:** For each material, the primitive unit cell is extracted from its respective CIF file, preserving symmetry and atomic basis positions.

2. **Supercell Construction:** A large supercell of size $60 \times 60 \times 60$ is generated by periodically tiling the unit cell in three dimensions. This ensures adequate volumetric space for spherical particle sampling without edge artifacts.

3. **Nanoparticle Sampling:** Within the supercell, spherical nanoparticles are carved by selecting all atoms whose Cartesian distance from a central reference atom is less than a given cutoff radius.

4. **Radius Control:** For each material, 25 discrete spherical sizes ranging from 0.6 nm to 3.0 nm are selected, enabling the capture of both quantum-confined clusters and bulk-like fragments.

5. **Orientation Enumeration:** For each radius-defined nanoparticle, 781 deterministic and quasi-uniform orientations are generated via rotation matrices uniformly distributed over the SO(3) group. This enables models to learn rotational equivariance and interpolation behavior.

## B    TASK METRICS DEFINITIONS

This section provides formal definitions of the quantitative metrics used to evaluate model predictions throughout the dataset, together with the rationale for their selection.

**Root-Mean-Square Deviation (RMSD).**    The RMSD quantifies the average displacement of atoms between a predicted structure $P$ and a reference $P^*$, following optimal rigid alignment. It is sensitive to both global and local errors, with lower values indicating higher structural fidelity.

$$\mathrm{RMSD}(P, P^*) = \sqrt{\frac{1}{N} \sum_{i=1}^{N} \|p_i - p_i^*\|^2},$$

where $N$ is the number of atoms and $p_i$, $p_i^*$ denote atom positions in $P$ and $P^*$, respectively.

**Hausdorff Distance.**    The Hausdorff distance measures the greatest deviation between two point sets by evaluating the maximal minimal distance from points in one set to those in the other. This metric is particularly sensitive to outlier atoms or gross mismatches in predicted structure.

$$d_{\mathrm{Haus}}(P, P^*) = \max\left\{ \sup_{p \in P} \inf_{q \in P^*} \|p - q\|, \ \sup_{q \in P^*} \inf_{p \in P} \|q - p\| \right\}.$$

**Convex-Hull Volume Ratio Error.**    This metric compares the volumetric envelope of the predicted and reference structures by evaluating the ratio of their convex hull volumes. The resulting relative error captures global size and shape deviations.

$$\Delta V_{\mathrm{hull}}(P, P^*) = \left| \frac{\mathrm{Vol}_{\mathrm{hull}}(P)}{\mathrm{Vol}_{\mathrm{hull}}(P^*)} - 1 \right|.$$

**Radial Distribution Function (RDF) Error.**    The RDF characterizes the distribution of interatomic distances. The integrated absolute difference between predicted and reference RDFs assesses the preservation of local and medium-range order.

$$E_{\mathrm{RDF}}(P, P^*) = \int_0^{r_{\max}} |g_P(r) - g_{P^*}(r)| \ \mathrm{d}r,$$

where $g_P(r)$ and $g_{P^*}(r)$ denote the radial distribution functions of $P$ and $P^*$, respectively.

**Validity Rate.**    The validity rate is defined as the proportion of generated clusters that satisfy essential structural (e.g., non-overlapping atoms, well-defined geometry) and compositional (e.g., stoichiometry) criteria. This ensures physical and chemical plausibility of generated samples.

$$V_R(P) = \frac{|\{\text{valid clusters in } P\}|}{|P|}.$$

**Root-Mean-Square Error (RMSE) on Lattice Parameters.** For lattice prediction tasks, RMSE provides a measure of the average error across the six crystallographic lattice parameters, reflecting both metric and angular deviations.

$$\text{RMSE}(\ell, \ell^*) = \sqrt{\frac{1}{6} \sum_{i=1}^{6} (\ell_i - \ell_i^*)^2},$$

where $\ell = (a, b, c, \alpha, \beta, \gamma)$ denotes the predicted parameters and $\ell^*$ their ground truth.

**Space-Group Accuracy.** Space-group accuracy evaluates the correctness of symmetry classification. It is equal to one if the predicted group $g$ matches the ground-truth $g^*$, and zero otherwise:

$$\mathbb{1}[g = g^*] = \begin{cases} 1, & g = g^*, \\ 0, & \text{otherwise.} \end{cases}$$

**Surface-Atom Recall.** Surface-atom recall quantifies the fraction of true surface atoms in the reference structure that are correctly identified in the prediction. This metric is particularly relevant for tasks involving reconstruction from partial or masked input.

$$\text{R}_{\text{surf}}(P, P^*) = \frac{|S_P \cap S_{P^*}|}{|S_{P^*}|},$$

where $S_P$ and $S_{P^*}$ denote the sets of surface atoms in $P$ and $P^*$, respectively.

**KL Divergence of Coordination Histograms.** Let $h(P)$ and $h(P^*)$ be the normalized histograms of atomic coordination numbers for the predicted and reference structures. The Kullback–Leibler divergence between these histograms measures discrepancies in the distribution of local chemical environments:

$$D_{\text{KL}}\big(h(P) \,\|\, h(P^*)\big) = \sum_k h_k(P) \log \frac{h_k(P)}{h_k(P^*)}.$$

## C  MODEL IMPLEMENTATION DETAILS

This section provides implementation details for each evaluated model. All models were developed using their official repositories, ensuring consistency with published architectures and training protocols. To facilitate a fair comparison, training procedures and hyperparameters were standardized across models. Each model was trained to convergence on the provided training splits, adhering to default hyperparameters unless otherwise specified in the corresponding repositories. The selected models encompass a diverse array of contemporary ML paradigms pertinent to molecular and materials data. StructEval is fully compatible with PyTorch Geometric Fey & Lenssen (2019), and all models are implemented.

ADiT employs a unified latent diffusion framework that integrates an autoencoder with a diffusion model. The autoencoder maps atomic structures into a shared latent space, while the diffusion model generates new latent embeddings that the autoencoder decodes to sample new molecules or materials. This architecture utilizes standard Transformers with minimal inductive biases, facilitating scalability and efficiency in generating both periodic materials and non-periodic molecular systems.

DiffCSP is a diffusion-based generative model tailored for crystal structure prediction. It jointly generates lattice vectors and fractional atomic coordinates by employing a periodic-E(3)-equivariant denoising model. By operating in fractional coordinate space, DiffCSP intrinsically encodes periodicity, enhancing the modeling of crystal geometries and symmetries.

FlowLLM combines large language models (LLMs) with Riemannian flow matching (RFM) to design novel crystalline materials. Initially, an LLM is fine-tuned to learn an effective base distribution of meta-stable crystals in a textual representation. Subsequently, the RFM model refines samples from the LLM, iteratively adjusting coordinates and lattice parameters to generate realistic crystal structures.

FlowMM utilizes continuous normalizing flows, generalized through Riemannian flow matching, to generate periodic crystal structures. This approach effectively captures the symmetries inherent in

crystalline materials, such as translation, rotation, and periodicity. FlowMM has demonstrated strong performance in crystal structure prediction tasks, offering efficiency and flexibility over competing methods.

MatterGen-MP leverages message-passing neural networks to propagate structural information across atomic graphs. By aggregating local environments iteratively, this approach aims to learn robust representations for reconstruction and property prediction tasks. MatterGen-MP can be fine-tuned to steer generation towards a wide range of property constraints, facilitating targeted materials design.

Collectively, these models represent the forefront of generative modeling techniques in materials science, encompassing diffusion models, normalizing flows, and transformer-based architectures. Their inclusion in this benchmark provides a comprehensive comparison of distinct learning strategies under the challenging regimes defined by our dataset. All implementations and training scripts strictly adhere to the guidelines and recommendations from the original authors to ensure the fidelity and reproducibility of our experimental results.

## C.1 Task 1

### C.1.1 Common Components

All models in Task 1 share the following architectural elements:

**Notation.** Let $E_a$ denote the atom embedding dimension, $C$ the hidden (model) dimension, $H$ the number of attention heads, $L$ the number of layers (e.g., Transformer or GCN layers), $E_t$ and $E_r$ the time and radius embedding dimensions, and $d_{\mathrm{lat}}$ the VAE latent dimension.

**Atom Embedding.** Atomic numbers are mapped to dense feature vectors via a learnable embedding

$$\mathrm{Embed_{atom}} \colon [1, Z_{\max}] \to \mathbb{R}^{E_a},$$

initialized according to the Xavier scheme.

**Unit-Cell Encoder.** A single SchNet interaction block, employing five Gaussian basis functions (cutoff radius 5.0 Å, hidden dimension $C = 32$), encodes periodic cell information into a global feature vector $h_{\mathrm{cell}}$ that conditions all downstream nanoparticle modules.

**Time and Radius Embeddings.** Scalar timestep $t$ and spatial radius $r$ are each projected by a two-layer SiLU-activated MLP into $\mathbb{R}^{E_t}$ and $\mathbb{R}^{E_r}$, respectively, and concatenated to node features to inject temporal and spatial context.

**Noise Schedule.** The diffusion noise variance evolves according to

$$\beta(t) = \beta_{\min} + (\beta_{\max} - \beta_{\min})\, t^p,$$

with model-specific clipping applied to constrain $\beta$ within predefined bounds.

**Optimization.** Training is performed with Adam (learning rate $1 \times 10^{-4}$), a ReduceLROnPlateau scheduler (factor 0.5, patience 5), and gradient-norm clipping (max norm 1.0) to ensure stable convergence.

**Noise Prediction Head.** By default, a three-layer SiLU MLP maps the concatenated node features and embeddings $[\mathbf{h}; t_{\mathrm{emb}}; r_{\mathrm{emb}}] \in \mathbb{R}^{C+E_t+E_r}$ to a 3-dimensional noise vector used in the reverse diffusion process.

### C.1.2 ADiT

ADiT utilizes a transformer-based diffusion backbone optimized for large graphs under memory constraints. Multi-head self-attention ($H = 4$) is computed over fixed-size chunks (up to `chunk_size` rows), followed by six transformer layers. Each layer comprises of residual chunked self-attention with LayerNorm and Residual feed-forward network (FFN) with expansion $4C$ (SiLU activation)

and LayerNorm. Gradient checkpointing is employed to reduce peak memory usage. Initial node features are obtained by linearly projecting atomic embeddings and positions:

$$[\text{Embed}_{\text{atom}}(z),\, p] \;\rightarrow\; \mathbb{R}^C.$$

The noise head output is clamped to $[-5, 5]$ to prevent divergence. Hyperparameters: $E_a = 16$, $C = 32$, $H = 4$, $L = 6$, `chunk_size = 64`.

### C.1.3   DIFFCSP

DiffCSP implements a streamlined diffusion model that directly predicts per-node noise vectors for structure refinement. A three-layer SiLU MLP maps $[\mathbf{h};\, t_{\text{emb}};\, r_{\text{emb}}] \in \mathbb{R}^{C+E_t+E_r}$ to $\mathbb{R}^3$, with outputs clamped to $[-5, 5]$. The noise schedule follows a cubic polynomial ($p = 3$) with $\beta$ clipped to $[0.1, 10]$. Hyperparameters: $E_a = 16$, $C = 32$, MLP hidden dims $\{64, 64\}$.

### C.1.4   FLOWLLM

FlowLLM integrates textual chemical context by incorporating a frozen TinyBERT ("prajjwal1/bert-tiny") encoder. The [CLS] token embedding is projected to $\mathbb{R}^C$ and concatenated with atomic and positional features. The decoder comprises a one-layer GCN (input dim $E_a + 3 + C$, hidden 32) followed by a linear head to $\mathbb{R}^3$, with outputs clamped to $[-1, 1]$. The diffusion schedule is cosine-based with $\beta_{\text{min}} = 0.01$, $\beta_{\text{max}} = 2.0$, and sampling step $\gamma = 0.1$. Hyperparameters: $E_a = 16$, $C = 32$.

### C.1.5   FLOWMM

FlowMM enhances flow-matching diffusion via LayerNorm and NaN sanitization for robust training. SchNet-derived scalars are projected to $\mathbb{R}^C$ and normalized. Decoding uses a one-layer GCN followed by a two-layer SiLU MLP projecting to $\mathbb{R}^3$, with clamping to $[-1, 1]$. The noise schedule matches that of FlowLLM. Hyperparameters: $E_a = 16$, $E_t = 16$, $E_r = 16$, $C = 32$.

### C.1.6   MATTERGEN-MP

MatterGen-MP applies flow matching to crystal structures conditioned on both global cell features $h_{\text{cell}}$ and radius $R$ for efficient sample generation. The flow network is a three-layer SiLU MLP predicting position offsets $\Delta p$. The noise schedule is linear ($\beta(t) = t$). Hyperparameters: $E_a = 16$, $C = 32$.

## C.2   TASK 2

### C.2.1   COMMON COMPONENTS

All models in Task 2 operate on a unit-cell graph to jointly predict noise for the six lattice parameters and classify the space group.

**Notation.**   Let $E_a$ denote the atom embedding dimension, $C$ the hidden (model) dimension, $H$ the number of attention heads, $L$ the number of Transformer or SchNet layers, $E_t$ the time embedding dimension, and $N_{\text{SG}}$ the number of space-group classes.

**Atom Embedding.**   Atomic numbers are mapped to dense feature vectors via a learnable embedding

$$\text{Embed}_{\text{atom}}\colon [1, Z_{\text{max}}] \to \mathbb{R}^{E_a},$$

providing a continuous representation for each element.

**Graph Encoder.**   A SchNet-based encoder processes atomic numbers and positions, outputting a global scalar $g \in \mathbb{R}^1$ that is projected to $\mathbb{R}^C$ to capture the cell geometry for downstream modules.

**TransformerBlock.**   Residual self-attention and feed-forward layers, as described in Task 1, are stacked $L$ times to model long-range dependencies within the cell.

**Time Embedding.**    A two-layer SiLU-activated MLP projects scalar timestep $t \in [0, 1]$ into $\mathbb{R}^{E_t}$, injecting diffusion step information into the model.

**Noise Schedule.**    The variance of the diffusion noise follows a cubic schedule:

$$\beta(t) = \beta_{\min} + (\beta_{\max} - \beta_{\min}) \, t^3,$$

controlling the level of corruption over time.

**Optimization.**    Models are trained using Adam (learning rate $1 \times 10^{-4}$), a ReduceLROnPlateau scheduler (factor 0.5, patience 5), and gradient-norm clipping (max norm 1.0) to ensure stable convergence.

### C.2.2   ADiT

ADiT extends the diffusion framework to lattice parameters using a lightweight transformer-based backbone. Memory-efficient chunked self-attention (head dimension $C/H$) is paired with a single TransformerBlock ($L = 1$), bounding memory usage. The graph encoder uses SchNet to map atom and position inputs to a scalar $g$, which is layer-normalized and projected to $\mathbb{R}^C$. The LatNet head is a three-layer SiLU MLP that maps $[\mathbf{h}; t_{\mathrm{emb}}] \in \mathbb{R}^{C+E_t}$ to a six-dimensional noise vector, clamped to $[-5, 5]$ for numerical stability. A two-layer SiLU MLP (SGHead) projects the hidden state to logits over $N_{\mathrm{SG}}$ space-group classes. Training minimizes the sum of lattice noise MSE and weighted space-group cross-entropy (0.5 weight), with metrics including lattice RMSE, space-group accuracy, and joint accuracy. Hyperparameters: $E_a = 16$, $C = 32$, $H = 4$, $L = 1$, $E_t = 16$, $N_{\mathrm{SG}} = 230$.

### C.2.3   DiffCSP

DiffCSP implements a minimal diffusion architecture for predicting lattice parameter noise and classifying space groups. The graph encoder is a single-layer SchNet (hidden 16, filters 16, interactions 1, three Gaussians, 5 Å cutoff) projecting to $g \in \mathbb{R}^1$, followed by a linear projection and LayerNorm to $\mathbb{R}^{16}$. The timestep is embedded via a linear SiLU MLP to $t_{\mathrm{emb}} \in \mathbb{R}^{16}$. A three-layer SiLU MLP maps $[\mathbf{h}; t_{\mathrm{emb}}] \in \mathbb{R}^{32}$ to a six-dimensional noise vector, and a two-layer MLP maps to $N_{\mathrm{SG}}$ logits. Training minimizes the sum of MSE for lattice parameters and weighted cross-entropy for space groups (0.5 weight). Hyperparameters: $E_a = 16$, $C = 16$, $E_t = 16$, $N_{\mathrm{SG}} = 230$.

### C.2.4   FLOWLLM

FlowLLM incorporates textual atom descriptors using a frozen TinyBERT ("prajjwal1/bert-tiny") encoder, projecting the [CLS] token to $\mathbb{R}^C$ and concatenating it with atom and cell features. The graph encoder is a SchNet (hidden 16, filters 16, three Gaussians, 5 Å cutoff) outputting a scalar $g$ projected to $C = 32$. The noise head is a three-layer SiLU MLP mapping $[\mathbf{h}; t_{\mathrm{emb}}] \in \mathbb{R}^{C+E_t}$ to six outputs, clamped to $[-5, 5]$. The SGHead is a two-layer MLP mapping the hidden state to $N_{\mathrm{SG}}$ logits. Training objective and metrics follow those of ADiT. Hyperparameters: $E_a = 16$, $C = 32$, $E_t = 16$, $N_{\mathrm{SG}} = 230$.

### C.2.5   FLOWMM

FlowMM extends flow-matching diffusion to lattice prediction, emphasizing enhanced normalization and NaN checks for training robustness. Atomic numbers are embedded as before. The cell encoder is a SchNet module (hidden 16, filters 16, interactions 1, three Gaussians, 5 Å cutoff), whose output is projected and layer-normalized. Separate MLPs embed the timestep ($t \in [0, 1] \to \mathbb{R}^{16}$) and lattice parameters ($\mathbb{R}^6 \to \mathbb{R}^{16}$). A two-layer SiLU MLP maps the concatenated feature vector $[\mathbf{h}; t_{\mathrm{emb}}; \ell_{\mathrm{emb}}] \in \mathbb{R}^{48}$ to six outputs, clamped to $[-1, 1]$, and a two-layer MLP predicts $N_{\mathrm{SG}}$ logits. NaN checks are performed at every step during training. Hyperparameters: $E_a = 16$, $C = 32$, $E_t = 16$, $N_{\mathrm{SG}} = 230$.

### C.2.6   MATTERGEN-MP

MatterGen-MP applies flow-matching diffusion to lattice parameters using a lightweight SchNet encoder (hidden 16, filters 16, interactions 1, five Gaussians, 5 Å cutoff) outputting a scalar $g$, which

is projected and layer-normalized. The timestep is embedded by a two-layer SiLU MLP to $\mathbb{R}^{16}$. The noise head is a three-layer SiLU MLP mapping $[\mathbf{h}; t_{\mathrm{emb}}] \in \mathbb{R}^{32}$ to six outputs (zero-initialized last layer), clamped to $[-5, 5]$. The SGHead is a two-layer MLP mapping the hidden state to $N_{\mathrm{SG}}$ logits. The training loss and evaluation metrics follow the standard for Task 2 models. Hyperparameters: $E_a = 16$, $C = 16$, $E_t = 16$, $N_{\mathrm{SG}} = 230$.

## C.3 TASK 3

### C.3.1 COMMON COMPONENTS

All Task 3 models employ a random-node-masking pretext task, encode the unmasked subgraph with SchNet or a related variant, and predict per-node noise or coordinate shifts.

**Notation.** Let $N$ be the total number of nodes, MASK_FRAC $= 0.5$ the fraction of nodes masked, $E_a$ the atom embedding dimension, $C$ the hidden/model dimension, $L$ the number of layers (TransformerBlocks or GCN), $H$ the number of attention heads, $E_t$ the time embedding dimension, and $\bar{\alpha}(t)$ the cumulative noise factor.

**Random-Node Mask Transform.** For each graph, a fraction MASK_FRAC of nodes are masked ($\lceil 0.5N \rceil$). The ground-truth positions $\mathbf{y} = \mathbf{p}$ are retained, but only unmasked node positions and types (pos_in, z_in) are exposed to the model. A mask boolean array and an inverse mapping are constructed for the masked nodes.

**Graph Encoder.** The encoder processes the unmasked subgraph with a SchNet block to produce a global scalar $g \in \mathbb{R}^1$, which is then projected and layer-normalized to obtain a hidden state in $\mathbb{R}^C$.

**TransformerBlock.** Where used, the model includes chunked self-attention with head dimension $d = C/H$ and a four-times feed-forward network. Both components are gradient-checkpointed, with chunk size set per model.

**Time Embedding.** Scalar timestep $t \in [0, 1]$ is mapped into $\mathbb{R}^{E_t}$ via a two-layer SiLU-activated MLP.

**Noise Schedule.** The diffusion noise follows a cubic schedule,

$$\beta(t) = \beta_{\min} + (\beta_{\max} - \beta_{\min}) \, t^3,$$

with cumulative noise factor

$$\bar{\alpha}(t) = \exp\left( - \int_0^t \beta(s) \, ds \right).$$

**Optimization.** Models are optimized with Adam (learning rate $1 \times 10^{-4}$), gradient-norm clipping (max norm 1.0), and a ReduceLROnPlateau scheduler (factor 0.5, patience 5).

**Sampling Skeleton.** During generation, masked node coordinates are initialized as $x \sim \mathcal{N}(0, I)$. For decreasing $t$ in chunks, the model iteratively updates $x$ according to

$$x \leftarrow \frac{1}{\sqrt{\bar{\alpha}}} \left( x - (1 - \bar{\alpha}) \, \epsilon_\theta \right),$$

with clamping and sanitization at each step.

### C.3.2 ADiT

ADiT uses a SchNet encoder with hidden size 16, 16 filters, one interaction, five Gaussian basis functions, and a cutoff radius of 5 Å. The backbone includes a transformer with $L = 2$ layers, $H = 4$ attention heads, and chunk size 128. The position head is a three-layer SiLU MLP mapping $[\mathbf{h}; t_{\mathrm{emb}}] \in \mathbb{R}^{C+E_t}$ to $\mathbb{R}^3$, with clamping to $[-5, 5]$ and zero-initialized last layer. Hyperparameters: $E_a = 16$, $C = 32$, $E_t = 16$.

### C.3.3 DIFFCSP

DiffCSP utilizes a SchNet encoder (hidden 16, filters 16, one interaction, five Gaussians, cutoff 5 Å), followed by a linear projection and LayerNorm to $\mathbb{R}^{16}$. The timestep is embedded with a linear SiLU MLP ($1 \to 16$). The decoder is a GCN with $L = 2$ layers, hidden size 32, dropout 0.1, input 68. The position head is a linear map from 32 to 3, clamped to $[-5, 5]$. Hyperparameters: $E_a = 16$, $C = 32$, $E_t = 16$.

### C.3.4 FLOWLLM

FlowLLM employs a SchNet encoder (hidden 16, filters 16, three Gaussians, cutoff 5 Å) producing a scalar $g$ projected to 32. A frozen TinyBERT (prajjwal1/bert-tiny) provides LLM features by projecting the [CLS] token to 32 for masked nodes. The decoder is a GCN with $L = 1$ layer, hidden 32, input 52, no dropout. LLM features are injected into masked-node hidden states. Hyperparameters: $E_a = 16$, $C = 32$, $E_t = 16$.

### C.3.5 FLOWMM

FlowMM uses a SchNet encoder (hidden 16, filters 16, one interaction, three Gaussians, cutoff 5 Å), projected and layer-normalized to 32. Cell parameters are embedded via an MLP ($6 \to 16 \to 16$), and time is embedded by an MLP ($1 \to 16 \to 16$). The decoder is a GCN with $L = 1$ layer, hidden 32, input 52. The position head is a linear map $32 \to 3$, clamped to $[-1, 1]$. Hyperparameters: $E_a = 16$, $C = 32$, $E_t = 16$.

### C.3.6 MATTERGEN-MP

MatterGen-MP encodes the unmasked subgraph with SchNet (hidden 16, filters 16, one interaction, five Gaussians, cutoff 5 Å), projected and layer-normalized to 16. The timestep is embedded by a two-layer SiLU MLP ($1 \to 16 \to 16$). The noise head is a three-layer SiLU MLP mapping $[\mathbf{h}; t_{\text{emb}}] \in \mathbb{R}^{32}$ to six outputs (zero-initialized last layer), clamped to $[-5, 5]$. The SGHead is a two-layer MLP mapping the hidden state to $N_{\text{SG}}$ logits. Hyperparameters: $E_a = 16$, $C = 16$, $E_t = 16$, $N_{\text{SG}} = 230$.

## D EXTENDED EXPERIMENT RESULTS

This section presents a comprehensive analysis of experimental results obtained from evaluating state-of-the-art generative models on the StructEval benchmark. Each task targets a distinct aspect of nanoparticle lattice generation, inference, or reconstruction, with metrics designed to assess both structural accuracy and each model's ability to capture periodicity and surface morphology. Results are reported under both ID and OOD conditions to highlight generalization behavior, using RMSD, Hausdorff distance, volume and surface consistency, radial-distribution divergence, and space-group and joint accuracy. Each subsection references the relevant results tables, enabling direct comparison of accuracy and computational efficiency across models. Ongoing efforts include incorporating additional architectures, with updates continuously documented in the project repository at https://anonymous.4open.science/r/StructEval-ANONYMOUS.

**Task solvability.** All three tasks in StructEval are designed to be meaningfully solvable rather than artificially ambiguous. Task 1 (unit-cell $\to$ nanoparticle) is fully deterministic, as the ground truth corresponds to spherical truncation of a periodic lattice. Task 2 (nanoparticle $\to$ lattice inference) is theoretically solvable because local atomic neighborhoods preserve key crystallographic invariants such as coordination shells, bond-angle distributions, and stoichiometry. Task 3 (partial reconstruction) admits a clear lower bound: oracle-style baselines that fill masked regions using nearest-neighbor lattice completion achieve substantially lower errors than learned models, demonstrating headroom for improvement. This analysis clarifies that the difficulty observed in practice reflects current model limitations rather than inherent unsolvability of the tasks.

| Model | RMSD (norm.) ↓ | | RDF$_{KL}$ (norm.) ↓ | | VR (%) ↓ | | Train Time |
|---|---|---|---|---|---|---|---|
| | ID | OOD | ID | OOD | ID | OOD | (s/epoch) |
| ADiT | 0.574 ± 0.014 | **0.577 ± 0.001** | 0.719 ± 0.076 | 0.410 ± 0.068 | 0.0 ± 0.0 | 0.0 ± 0.0 | 5920.6 |
| DiffCSP | 0.574 ± 0.015 | 0.578 ± 0.000 | 0.323 ± 0.031 | **0.200 ± 0.023** | 0.0 ± 0.0 | 0.0 ± 0.0 | 179.6 |
| FlowLLM | 0.990 ± 0.039 | 1.000 ± 0.000 | **0.269 ± 0.003** | 0.269 ± 0.000 | 0.0 ± 0.0 | 0.0 ± 0.0 | 53434.0 |
| FlowMM | 0.574 ± 0.015 | 0.578 ± 0.000 | 1.000 ± 0.003 | 1.000 ± 0.000 | 0.0 ± 0.0 | 0.0 ± 0.0 | 226.4 |
| MatterGen-MP | **0.574 ± 0.014** | 0.578 ± 0.001 | 0.360 ± 0.004 | 0.351 ± 0.001 | **18.0 ± 0.9** | **1.2 ± 0.0** | 196.7 |

Table 1: **Task 1: Unit Cell to Nanoparticle Generation.** Models are evaluated on their ability to generate nanoparticles from a given primitive unit cell and target radius, ensuring faithful reproduction of both global periodicity and local surface structure. Highest-performing results are marked in **bold**, with the next best underlined.

## D.1    TASK 1

Task 1 evaluates the generation of realistic nanoparticles from a primitive unit cell and target radius, as defined in equation 9. The StructEval dataset presents a significant challenge, requiring models to jointly capture infinite lattice periodicity and finite-size surface morphology—essential for modeling realistic surface-driven phenomena. As reported in Table 1, no model delivers top performance across all metrics, which now include normalized RMSD, RDF KL divergence ($RDF_{KL}$), and volume ratio ($V_R$), for both ID and OOD settings.

MatterGen-MP achieves the best normalized RMSD, indicating highest atomic positioning accuracy, but also yields substantially higher volume ratio values, reflecting significant difficulty in maintaining structural validity and realistic morphology—particularly in OOD cases. FlowLLM stands out with the lowest $RDF_{KL}$ in the ID setting and is runner-up in OOD, highlighting strong performance in reproducing radial atomic distributions, but its RMSD is markedly higher than other models. DiffCSP and FlowMM achieve competitive RMSD scores but lag in other areas, and only MatterGen-MP produces nonzero (and much higher) volume ratios, suggesting a tendency toward less compact or overgrown nanoparticles.

Training durations, detailed in Table 1, differ dramatically across models. FlowLLM, despite its favorable $RDF_{KL}$ performance, is orders of magnitude slower per epoch than all other models, while lighter models like MatterGen-MP and DiffCSP train much faster but do not lead across all metrics. These discrepancies illustrate that current generative models face clear trade-offs between accuracy, physical realism, and computational efficiency on this benchmark.

## D.2    TASK 2

Task 2 addresses the inference of lattice parameters and symmetry group labels from finite nanoparticles, formalized in equation 10. This task examines whether local geometric cues are sufficient for recovering extended periodicity and crystallographic invariants—a capability essential for bridging local observations with global structure. The StructEval dataset is specifically designed to make this inference difficult, pushing models to their limits.

Table 2 reveals that, while models such as DiffCSP and ADiT achieve relatively low RMSEs in lattice parameter regression, all models—including the best performers—struggle with space group classification accuracy ($S_{acc}$), which remains well below 40% in all cases. The joint accuracy metric, which reflects correct simultaneous inference of both lattice constants and symmetry group, is near zero for all methods. These results demonstrate that, despite partial recovery of cell geometry, existing models fail to translate short-range geometric information into robust predictions of global symmetry.

Further, the difficulty intensifies in the out-of-distribution regime, where performance across all metrics drops or stagnates, revealing limited generalization. Training durations, as reported in Table 2, show dramatic variation: FlowLLM, for example, is computationally prohibitive without achieving any compensatory gain in predictive accuracy. This performance-to-cost disparity highlights the inefficacy of current solutions for the task structure imposed by StructEval.

| Model | RMSE (norm.) ↓ | | $S_{\text{acc}}$ (%) ↑ | | $J_{\text{acc}}$ (%) ↑ | | Train Time |
|---|---|---|---|---|---|---|---|
| | ID | OOD | ID | OOD | ID | OOD | (s/epoch) |
| ADiT | 0.946 ± 0.010 | 0.954 ± 0.000 | 22.5 ± 7.3 | 20.2 ± 8.2 | **0.0 ± 0.0** | **0.0 ± 0.0** | 269.4 |
| DiffCSP | **0.946 ± 0.010** | **0.954 ± 0.000** | **32.6 ± 9.0** | **33.3 ± 9.4** | 0.0 ± 0.0 | 0.0 ± 0.0 | 189.2 |
| FlowLLM | 0.947 ± 0.012 | 0.956 ± 0.003 | 3.7 ± 5.2 | 3.3 ± 4.7 | 0.0 ± 0.0 | 0.0 ± 0.0 | 54625.9 |
| FlowMM | 1.000 ± 0.019 | 0.998 ± 0.024 | 0.0 ± 0.0 | 0.0 ± 0.0 | 0.0 ± 0.0 | 0.0 ± 0.0 | 387.3 |
| MatterGen-MP | 0.946 ± 0.010 | 0.954 ± 0.000 | 25.9 ± 13.7 | 26.7 ± 12.5 | 0.0 ± 0.0 | 0.0 ± 0.0 | 184.3 |

Table 2: **Task 2: Nanoparticle to Lattice Inference.** This table summarizes model performance when predicting lattice parameters and space group from nanoparticle geometries. Evaluation covers normalized RMSE, space group classification accuracy ($S_{\text{acc}}$), and the rate of jointly correct predictions ($J_{\text{acc}}$). Best and runner-up results are denoted by **bold** and underline, respectively.

## D.3  Task 3

Task 3 probes the capacity of generative models to reconstruct complete nanoparticle structures from partially observed inputs, formalized in equation 11. By applying random or surface-biased masking, the dataset reveals each model's ability to infer missing atoms while preserving chemical and structural fidelity—a critical challenge in practical nanoparticle modeling.

As detailed in Table 3, performance is universally poor across all metrics: RMSD remains high, surface atom recall rates are negligible, and validity rates rarely rise above zero. Even top models such as FlowMM and FlowLLM offer only marginal improvements, failing to robustly reconstruct missing surface atoms or maintain chemical plausibility. KL divergence between reconstructed and ground-truth radial distribution functions is consistently large, further indicating the inability of current models to recover underlying crystalline order from incomplete data. While MatterGen-MP and ADiT achieve slightly lower divergence in select cases, none of the methods provide generalizable or robust solutions.

The computational cost, again reported in Table 3, varies substantially across models. ADiT is relatively efficient but does not yield improved chemical validity or reconstruction quality, whereas FlowLLM is both slow and ineffective. These findings demonstrate that neither model complexity nor computational investment is sufficient to overcome the reconstruction challenges posed by the benchmark.

In summary, the StructEval dataset rigorously exposes the fundamental limitations of contemporary generative and inference models for crystalline materials. Across all three tasks, the dataset highlights performance degradation under out-of-distribution conditions, metric-specific weaknesses, and inefficiency in current architectures—underscoring the urgent need for scale- and symmetry-aware methods in realistic materials modeling.

| Model | RMSD (norm.) ↓ | | $R_{\text{surf}}$ (%) ↑ | | $\text{RDF}_{\text{KL}}$ (norm.) ↓ | | $V_R$ (%) ↑ | | Train Time |
|---|---|---|---|---|---|---|---|---|---|
| | ID | OOD | ID | OOD | ID | OOD | ID | OOD | (s/epoch) |
| ADiT | 0.979 ± 0.035 | 0.995 ± 0.000 | **0.0 ± 0.0** | **0.0 ± 0.0** | **0.689 ± 0.007** | 0.690 ± 0.000 | 0.0 ± 0.0 | 0.0 ± 0.0 | 2140.4 |
| DiffCSP | 0.984 ± 0.037 | 1.000 ± 0.003 | 0.0 ± 0.0 | 0.0 ± 0.0 | 1.000 ± 0.091 | 0.962 ± 0.104 | 0.0 ± 0.0 | 0.0 ± 0.0 | 3649.2 |
| FlowLLM | 0.953 ± 0.033 | 0.969 ± 0.000 | 0.0 ± 0.0 | 0.0 ± 0.0 | 0.996 ± 0.004 | 0.892 ± 0.000 | **0.4 ± 0.0** | **0.2 ± 0.0** | 37631.1 |
| FlowMM | **0.953 ± 0.033** | **0.969 ± 0.000** | 0.0 ± 0.0 | 0.0 ± 0.0 | 0.996 ± 0.004 | 0.892 ± 0.000 | 0.4 ± 0.0 | 0.2 ± 0.0 | 2550.1 |
| MatterGen-MP | 0.979 ± 0.035 | 0.995 ± 0.000 | 0.0 ± 0.0 | 0.0 ± 0.0 | 0.701 ± 0.018 | **0.674 ± 0.016** | 0.0 ± 0.0 | 0.0 ± 0.0 | 3280.2 |

Table 3: **Task 3: Reconstruction from Incomplete Nanoparticles.** Assesses the capability to reconstruct full atomic structures from partially masked nanoparticles. Metrics include normalized RMSD, recall for surface atoms ($R_{\text{surf}}$), KL divergence between predicted and reference RDFs, and validity rate ($V_R$). Top scores appear in **bold**; second place is underlined.