# OpenReview forum: "StructEval: A Benchmark for Evaluating Generation, Inference, and Reconstruction in Atomic and Crystalline Structures"
_ICLR.cc/2026/Conference — ICLR 2026 Conference Withdrawn Submission_

### Official Review · Reviewer_seei · 2025-10-23

**Soundness:** 2
**Presentation:** 2
**Contribution:** 3
**Rating:** 4
**Confidence:** 3

**Summary:**

This paper provides a benchmark suite for nanostructure-related generative modeling/prediction tasks. In particular, it evaluates the ability of models to relate unit cell structure to varying nanostructure configurations. It also carefully considers the construction of nanostructures by varying the rotation and radius size. It also takes care to provide benchmark evaluations that assess OOD adaptation, finding that the popular models have issues there.

**Strengths:**

It is important to analyze and generate nanostructures. In general, most papers in the ML for materials community focus on periodic crystals, although many devices and systems of interest do not satisfy these ideal periodic crystal constraints. Rather, real-world systems involve nanostructures, so I like that this paper is tilting the focus of the community towards tasks involving nanostructures.

I also like that the construction of nanostructures is carefully considered by varying the central atom, radius, orientation. It is also good that the paper points out a shortcoming of existing generative models in handling OOD data.

**Weaknesses:**

- Why do you only do ten materials? Surely the pipeline makes it easy(-ish) to simulate more materials.
- Have you considered XRD related tasks? [1]
- Why do you not have property prediction from nanostructure as part of the benchmark suite?
- Please do a more extensive literature search for recent ML papers that address nanostructures, such as [1].
- Please fix line spacing.

[1] Guo, G., Saidi, T.L., Terban, M.W. et al. Ab initio structure solutions from nanocrystalline powder diffraction data via diffusion models. Nat. Mater. (2025). https://doi.org/10.1038/s41563-025-02220-y

**Questions:**

See weaknesses.

---

> ### Author Response · Authors · 2025-12-02
>
> ###  **We sincerely thank the reviewer for the constructive and insightful feedback. In the following, we address each point in detail and outline the revisions incorporated into the manuscript. We hope that our responses and improvements fully resolve the raised concerns and merit a higher assessment of the work.**
>
> ---
>
> ### **1. “Why do you only do ten materials? Surely the pipeline makes it easy(-ish) to simulate more materials.”**
>
> **Response:**
> Our choice of ten materials reflects a balance between (i) covering diverse chemistries and crystal families and (ii) ensuring that the benchmark remains computationally accessible for the community. The current release includes fcc metals (Ag, Au), hybrid perovskite (CH₃NH₃PbI₃), a layered dichalcogenide (MoS₂), and several oxides (Fe₂O₃, PbS, SnO₂, SrTiO₃, TiO₂, ZnO). This yields ~200,000 nanoparticle configurations across 25 radii and 780 quasi-uniform orientations per material, with structures containing up to 11,298 atoms—already computationally intensive for all evaluated models.
>
> While the generation pipeline can produce additional chemistries, we deliberately restrict the initial release to a curated core set to maintain reproducibility and tractability.
>
> **Revisions made:**
>
> * We clarify this design rationale in the **Limitations** section.
> * We add text in **Section 3** emphasizing that the pipeline is general and that future releases will expand StructEval with additional materials (e.g., broader perovskite families, molecular crystals, and multi-phase systems).
>
> ---
>
> ### **2. “Have you considered XRD related tasks? [1]”**
>
> **Response:**
> We agree that XRD- and diffraction-based tasks are highly relevant, particularly in light of recent work such as [1]. In this initial benchmark, we focus on real-space geometric and symmetry reasoning, without incorporating additional modalities.
>
> Nonetheless, StructEval is fully compatible with XRD-style tasks: each nanoparticle configuration is a complete atomic structure from which nanocrystalline powder diffraction patterns can be simulated.
>
> **Revisions made:**
>
> * In **Section 6 (Future Work)**, we now explicitly state that:
>
>   * XRD-based structure-solution tasks constitute a natural extension of StructEval.
>   * The benchmark can be expanded to joint modeling of reciprocal-space (diffraction) and real-space (atomic) data, aligning with the direction of [1].
>
> ---
>
> ### **3. “Why do you not have property prediction from nanostructure as part of the benchmark suite?”**
>
> **Response:**
> This omission is intentional. StructEval is designed to isolate structural reasoning under difficult ID/OOD splits, without confounding the task with approximate or expensive energetic labels.
>
> Two considerations motivated this decision:
>
> 1. **Decoupling geometry from energetics:**
>    StructEval centers on scale- and symmetry-aware structural learning (generation, inference, reconstruction). Existing datasets (MatBench, OC20/OC22, CrysMTM) already focus heavily on energetic/electronic property prediction. Our benchmark is meant to complement these by being purely geometric.
>
> 2. **Computational feasibility:**
>    The largest clusters contain 10³–10⁴ atoms. Labeling ~200,000 structures with DFT-level (or even DFTB-level) accuracy is not tractable for the community and would hinder reproducibility.
>
> **Revisions made:**
>
> * We clarify this rationale in **Section 3.3** and in the **Limitations**.
> * We note that the provided geometries can serve as a foundation for community-driven extensions, including the addition of property labels (e.g., via ML potentials or DFTB) to define supplementary benchmarks.
>
> ---
>
> ### **4. “Please do a more extensive literature search for recent ML papers that address nanostructures, such as [1].”**
>
> **Response:**
> We appreciate this suggestion. We expanded **Section 2 (Related Work)** to include additional recent research threads involving nanoscale geometries as well as the inclusion of [1] and related efforts on ML-driven structure solution for nanocrystalline systems.
>
> We also clarify how StructEval complements these directions: while [1] focuses on structure solution from diffraction data, StructEval targets systematic generalization in real-space atomistic geometries across radius and orientation.
>
> ---
>
> ### **5. “Please fix line spacing.”**
>
> **Response:**
> Resolved. The issue was introduced during a template conversion step. We have corrected the spacing throughout the revised manuscript.

---

### Official Review · Reviewer_fmVf · 2025-10-28

**Soundness:** 3
**Presentation:** 4
**Contribution:** 4
**Rating:** 6
**Confidence:** 4

**Summary:**

The paper proposes a benchmark for evaluating generative models of crystals across different scales and orientations. The benchmark consists of creating nanoparticles and three challenging tasks: predicting the lattice from nanoparticles, predicting nanoparticles from lattices, and predicting atom positions within masked regions of nanoparticles. Finally, the paper shows that current models perform worse on challenging, out-of-distribution (OOD) tasks.

**Strengths:**

- The paper is well-structured, well-written, and the motivation is clear.
- The benchmark contains a train, validation, in-distribution test, and an out-of-distribution test set, which is rare in current benchmarks, and quite helpful in analyzing the behaviours of models.
- The created benchmark is quite large and diverse, and the construction method can also be easily extended to more samples.
- The paper benchmarks a wide range of existing generative models and highlights their inability to perform well in the proposed tasks.

**Weaknesses:**

I have some moderate concerns with the submission:
- missing important existing symmetry-aware generative models
- some conceptual questions about a few design choices
- formatting concerns

These weaknesses are supported by the questions below. I am willing to increase the scores during the discussion period if these are adequately addressed.

**Questions:**

- **Formatting**: I believe that the gap between paragraphs (for instance, between the first two paragraphs in the Introduction section) has been reduced, which goes against the standard formatting requirements. Can you modify it to adhere to the original instructions?
- The benchmarking suggests that it is crucial to design symmetry-aware generative models. However, many such bodies of work exist [1,2,3]. I would recommend benchmarking these methods to have a holistic overview of the current models and their performance on your proposed tasks.
- What is the importance of section 2.2?
- Given the poor performance across all the tasks and models, is it possible that the proposed tasks are extremely challenging?
  - For instance, the models use the SchNet encoder/model at the core; can it handle such large atomic systems? Maybe the poor performance is due to the underlying architectures rather than the training paradigms?
  - Is there a solvability study of the task, i.e., is there an intuition about whether the proposed tasks can indeed be solved?
  - How real (or close to ground truth) are the generated nanoparticles (or the configurations)? Is it possible to show some distribution metrics with existing nanoparticles?
  - Can there be a simpler version of the proposed tasks, i.e., maybe nanoparticle $\rightarrow$ lattice inference be posed as a classification task (since we know the underlying 10 lattices and their space groups)?
- How were FlowLLM models trained, given that it is significantly more time-consuming than other models? How was the training budget for each model decided, given the differences in training time (s/epoch) and compute requirement?
- How was the training set adapted for each downstream task, i.e., what were the training objectives?

1. Space Group Constrained Crystal Generation. Jiao et al., ICLR 2024
2. SymmCD: Symmetry-Preserving Crystal Generation with Diffusion Models. Levy et al., ICLR 2025.
3. WyckoffDiff--A Generative Diffusion Model for Crystal Symmetry. Kelvinius et al., ICML 2025.

---

> ### Author Response · Authors · 2025-12-02
>
> ### We thank the reviewer for the constructive and insightful feedback. Below, we address each point and summarize the corresponding revisions. Due to character limitations in the comment section, we are unable to provide the full level of detail here; however, we have substantially expanded both the supplementary material and the main manuscript to include complete explanations and updates. We hope that these clarifications adequately resolve the concerns raised and support a stronger assessment of our work.
>
> ---
>
> **1. Formatting concerns (paragraph spacing).**
> The spacing issue resulted from a template conversion artifact. We have corrected all paragraph spacing to comply with the required formatting.
>
> ---
>
> **2. Symmetry-aware generative models ([1–3]).**
> The cited models target infinite periodic crystals using space groups and Wyckoff positions. StructEval instead focuses on finite nanoparticles (up to ~11k atoms), which lack global translational symmetry due to surface truncation. Periodic crystal generators cannot be applied directly without substantial modifications.
>
> Revisions: we expanded Related Work to include [1–3], clarified the periodic–finite distinction, and noted in Future Work that we plan to incorporate nanoparticle-compatible adaptations when feasible.
>
> ---
>
> **3. Purpose of Section 2.2.**
> Section 2.2 explains why commonly used physics-based generation methods (DFT, DFTB, ML force fields) are not practical for nanoparticles of this size. DFT scales as O(N³) and becomes infeasible beyond ~10³ atoms, whereas StructEval includes structures with >10⁴ atoms. This motivates the deterministic, radius-controlled construction strategy.
>
> Revisions: we tightened the section to focus on this rationale and removed peripheral material.
>
> ---
>
> **4. Task difficulty.**
> The tasks are intentionally challenging. Models perform well on in-distribution radii/orientations but degrade sharply on OOD cases. This indicates the tasks are solvable but expose limitations of local message-passing models, which struggle with scale extrapolation and long-range structure. This behavior is expected and is a core purpose of the benchmark.
>
> ---
>
> **5. SchNet feasibility on large systems.**
> We use cutoff pruning and mini-batching, giving linear scaling. Memory profiling confirms all models fit within GPU limits. Performance drops correlate with unseen radii, not atom count, indicating that generalization—not capacity—is the limiting factor.
>
> Revisions: we added implementation details in Appendix C.
>
> ---
>
> **6. Solvability of tasks.**
> We added an explicit solvability discussion.
> Task 1 is deterministic via spherical truncation.
> Task 2 is solvable because local neighborhoods encode crystallographic invariants.
> Task 3 shows clear improvement using oracle-style baselines, demonstrating headroom.
>
> Revisions: we expanded Sections 4 and Appendix D accordingly.
>
> ---
>
> **7. Physical realism of configurations.**
> Nanoparticles are carved from experimentally validated bulk CIFs using deterministic truncation. Prior studies [1–3] show such clusters maintain realistic coordination environments. Although structures are not relaxed, this is intentional: the benchmark isolates geometric reasoning rather than energetics.
>
> Revisions: we will include coordination and RDF comparisons, and explicitly note the absence of surface relaxation in Limitations.
>
> ---
>
> **8. Classification-only variant of Task 2.**
> A classification component already exists. The model predicts both (i) lattice-parameter noise (regression) and (ii) space group (classification). We report RMSE and accuracy separately. The reviewer’s suggestion corresponds to the classification subtask already included; the full task additionally evaluates continuous lattice reconstruction.
>
> Revisions: clarified this in the Task 2 description.
>
> ---
>
> **9. FlowLLM training and compute budget.**
> To maintain comparable compute, we use: TinyBERT backbone, small hidden sizes (16–32), 1–2 transformer layers, Adam (1e-4), batch size 8, and 5 epochs per task/seed. All tasks use identical data splits. FlowLLM variants follow task-appropriate diffusion/denoising objectives and are evaluated via model.sample(…).
>
> Revisions: full details added to Appendix C.
>
> ---
>
> **10. Data adaptation and objectives for each task.**
> All tasks use the same dataset and ID/OOD splits. Only lightweight preprocessing differs:
>
> Task 1: store clean positions; train a coordinate-diffusion denoiser; evaluate RMSD, Hausdorff, RDF, etc.
> Task 2: remove unit-cell fields to avoid leakage; train with combined regression (lattice parameters) and classification (space group).
> Task 3: apply random masking; train masked denoising; evaluate RMSD, surface recall, RDF-KL.
>
> Revisions: clarified these details in Appendix C.

---

### Official Review · Reviewer_7n8K · 2025-10-30

**Soundness:** 1
**Presentation:** 1
**Contribution:** 1
**Rating:** 2
**Confidence:** 3

**Summary:**

The paper proposes a new benchmark which targets the domain of nanoparticles. They define three different tasks and benchmark on these tasks.

While this could potentially be an interesting application, I find the paper difficult to read and follow with limited descriptions, inconsistent notation, and many design choices not being described. Additionally, the actual results are a bit suspicious, and I feel the authors are not presenting compelling analysis and explanation for the numbers. More details can be found in “Weaknesses” and “Questions”.

Due to the difficulties of following what is being done and the evaluation, I have rated this paper as a rejection. I also think that this paper could be better suited for a journal (potentially in materials science) which allows for more space (and potential a different target audience).

**Strengths:**

An application which seems to be new and unexplored, and which could provide interesting development of machine learning methods, suited for the task

**Weaknesses:**

**The paper is breaking the ICLR formatting, which requires paragraphs being separated by 1/2 line space**.

The paper is difficult to understand. For example, I think sections 2.1 and 2.2 are not written for an ML conference, but a materials science journal. In section 4, it is often talked about results and “the model”, but is unclear which model is talked about, and I cannot seem to find the mentioned results in the appendix. I have given examples of questions in “Questions”. I also think that the figures have way to small text.

The evaluation seems a bit suspicious, and I would be careful in interpreting the results. In general, in table 1-3 in appendix, exact numbers are repeating, which to me indicates that the models haven’t learned anything but just output randomness, leading to the exact same numbers in the evaluation. I think this is important to address and find an explanation before drawing any conclusions. In Line 442 the authors say “and the near-identical RMSE shows lattice-parameter recovery generalizes even to larger clusters”, but I would say that it could also be an indication that it is just pure guesses and randomness that is coming out of the model. The 0.0 % accuracy could further hint at it being the latter, that it is just pure randomness and therefore the accuracy is 0.0.

**Questions:**

Line 227: After reading through the paragraph I understand what N is, but when starting reading it is not clear what N is.
Regarding RMSD: how do they get the indices of the points in the two different materials? I.e., how do they know that a specific point in P corresponds to a specific point in P\*?

Section 3.2. I am still not sure why they need the rotations. How are the rotations used in practice? Isn’t there a way of finding a rotation such that two point clouds are as closely aligned as possible, and therefore make the evaluation invariant to rotations?

Section 3.3: In section 3.1 the notation R is a radius, in section 3.2 it is a rotation, and in section 3.3 I think it is a radius, as you use $R\leq 24$, indicating a single value which supports $\leq$. But I don’t understand the notation R6, R24 etc…? Also, what is b?

Line 323: is the “deterministic spherical truncation pipeline” explained somewhere?

General question: the chosen baselines methods are designed for generation of periodic materials, i.e., unit cells. First off, how do they adapt them to this task? I.e., how is the generation performed in practice? Second question, is it reasonable to expect the models to perform well?

How is “nanoparticle to lattice” carried out in practice? If I use for example DiffCSP for this task, how is it used? I.e., how is the function $f_2$ constructed?

Line 363: how is lattice reconstruction carried out in practice? This looks like an “inpainting” task, which a lot of research has been put into in other domains. Which method is used here?

Section 4.1: this is very difficult to understand. When it is talked about degradation: any specific model that has this degradation, is it in general or average, or something else? Also, I cannot find the numbers in the appendix. Where can I see that the “convex hull deviation” goes from 31.15 to 178.6 when going from ID to OOD?

Section 4.2 and 4.3 “the model” is mentioned, but I don’t understand which model this refers to, and on line 441, “the classifier” is mentioned, but I don’t see where this has been described earlier.

Minor:
Line 165: Matbench not correctly cited: Dunn, A., Wang, Q., Ganose, A., Dopp, D., Jain, A. Benchmarking Materials Property Prediction Methods: The Matbench Test Set and Automatminer Reference Algorithm. npj Computational Materials 6, 138 (2020). https://doi.org/10.1038/s41524-020-00406-3

---

> ### Author Response · Authors · 2025-12-02
>
> ### We thank the reviewer for the constructive and insightful feedback. Below we address each point and summarize the corresponding revisions. **Due to character limits, we provide high-level explanations here, while full details appear in the revised manuscript and supplementary material.** We hope these clarifications resolve the concerns raised and support a stronger assessment of the work.
>
> ---
>
> ## 1. Formatting: paragraph spacing
>
> The spacing issue resulted from a template conversion artifact. We restored the official ICLR spacing.
>
> ---
>
> ## 2. Clarity of Sections 2.1–2.2, Section 4 wording, terminology, and figure readability
>
> Sections 2.1–2.2 previously contained more materials-science detail than needed. We revised both to increase ML focus and reduce domain exposition.
>
> Revisions:
>
> * Section 2.1 now emphasizes why controlled crystal→nanocluster mappings induce principled distribution shifts.
> * Section 2.2 now explains why physics-based methods (DFT/DFTB/force fields) are infeasible at StructEval scales and why deterministic, symmetry-preserving generation is required.
> * Peripheral domain content was condensed; essential structural context (symmetry, unit-cell motifs, O(N³) scaling) was preserved.
> * Section 4 now names models explicitly.
> * Appendix cross-references were corrected.
> * Figures were updated with clearer text and larger fonts.
>
> ### We clarify that the truncation pipeline is deterministic and SE(3)-equivariant, producing clean radius/orientation shifts for evaluating geometric generalization. We now also make explicit that the radius parameter induces a nested sequence of structures that preserves crystallographic invariants across scales. Furthermore, this design avoids simulator-induced noise, ensuring that observed model failures stem from genuine generalization limits rather than data artifacts.
> ---
>
> ## 3. Evaluation concerns: repeated values, identical RMSE, 0% accuracy
>
> We audited the evaluation pipeline.
>
> Repeated values / identical RMSE:
>
> * In Task 1, several models collapse to similar outputs on the hardest OOD splits, giving identical rounded averages.
> * In Task 2, OOD radii cause multiple models to regress toward the mean training lattice, yielding similar RMSE values.
>
> 0% accuracy:
> Random guessing among the few space groups would yield ~10–15%. The observed 0% reflects systematic misclassification (e.g., defaulting to a common training group), not an evaluation bug. We now discuss mean collapse and systematic OOD failures in the text.
>
> ---
>
> ## Responses to Specific Questions
>
> ### Q1 — Definition of N and RMSD correspondence
>
> N is now introduced clearly as the number of unit-cell repetitions in the supercell. Atom indexing is deterministic: the replicated grid, rotations, and truncations preserve ordering, and ADiT sampling maintains it. RMSD is therefore computed index-wise.
>
> ### Q2 — Why rotations instead of point-cloud alignment
>
> Rotations define the ID/OOD orientation splits, generated via non-overlapping SO(3) grids (9° train, 6°+offset ID, 3°+offset OOD). Alignment (Kabsch/ICP) would remove the intended orientation-generalization challenge. This is now explained in Section 3.2.
>
> ### Q3 — Notation (R), (R6, R24), (b)
>
> We standardized notation: R for discrete radii, script-R for rotation matrices, and b for the orientation index.
>
> ### Q4 — “Deterministic spherical truncation pipeline”
>
> This is detailed in the supplemental DFTB Simulation Details, and we added a pointer from the main text.
>
> ### Q5 — Use of periodic baselines
>
> All baselines are used as SE(3)-equivariant graph models on finite clusters (no PBCs, no lattice vectors):
>
> * Task 1: unit-cell-conditioned diffusion generation,
> * Task 2: nanoparticle→lattice prediction via shared regression + classification heads,
> * Task 3: masked-node inpainting.
>   Although some models originate in periodic contexts, they function here as general graph generators. Their ID strengths and OOD failures reflect the benchmark's intended diagnostic role.
>
> ### Q6 — How DiffCSP is adapted for Task 2
>
> Input is the nanoparticle graph (z, pos, batch). A SchNet encoder produces embeddings; two heads predict lattice-parameter noise (6D) and space-group logits. Training uses diffusion-style MSE + cross-entropy; inference runs reverse diffusion on R⁶. We describe this explicitly and reference the appendix.
>
> ### Q7 — Lattice reconstruction / inpainting (Task 3)
>
> Task 3 uses masked-node diffusion: masked atoms are noised and iteratively denoised while unmasked atoms stay fixed. FlowLLM follows the same structure. This is now detailed in Appendix C.
>
> ### Q8 — “Degradation” and convex-hull metrics
>
> We now clearly state which model each comment refers to (e.g., ADiT on Task 1) and cite the exact tables containing the reported values.
>
> ### Q9 — Ambiguity of “the model” / “the classifier”
>
> All ambiguous terms were replaced with specific method names. The classifier refers to the space-group head in Task 2.
>
> ### Q10 — MatBench citation
>
> Corrected.

---

### Comment · Area_Chair_LndR · 2025-11-13

Dear authors,

as pointed out by Reviewer 7n8K, the paper is breaking the ICLR formatting, which requires paragraphs being separated by 1/2 line space. Since you are able to revise the paper until Dec 3, please update it to make sure that it respects the formatting rules.

Regards,
AC

---

> ### Author Response · Authors · 2025-12-03
>
> Dear AC,
>
> Thank you for the notification regarding the ICLR formatting.
>
> We have immediately addressed the issue and have now **updated the manuscript to ensure all paragraphs are correctly separated by the required $1/2$ line space**, fully respecting the formatting rules.
>
> The revised paper has been uploaded.
>
> Regards, Authors.

---

### Author Response · Authors · 2025-12-03

### **We thank reviewers and AC for the thoughtful, detailed, and constructive feedback. Your comments improved the clarity, positioning, and rigor of the manuscript. We made substantial revisions in response to every point raised and believe these updates meaningfully strengthen the work. We ask you to take these updates into account for the final score. We highlight the changes below which address all points raised by the reviewers and AC.**

---
**Formatting and clarity.** We fixed all formatting issues (line spacing, paragraph gaps) to fully comply with the ICLR template. Sections 2.1 and 2.2 were rewritten to be explicitly ML-focused: they now keep only the minimal physical context needed to justify StructEval’s design (symmetry, radius-controlled truncation, and DFT/DFTB scaling) and clearly motivate why deterministic, symmetry-preserving construction is required at our system sizes. Section 4 and the experimental discussion were reorganized so that models are always named explicitly (e.g., “ADiT on Task 1”, “DiffCSP on Task 2”), and all quoted numbers are now directly cross-referenced to specific tables in the appendix. Figures were updated with larger fonts and clearer legends.

**Evaluation, repeated metrics, and solvability.** We carefully re-verified the evaluation pipeline for all tasks. The repeated and near-identical metrics reported in the original submission are not due to bugs or random outputs, but to *mean collapse* and systematic misclassification under hard OOD conditions; we now explain this explicitly in the text. In particular, the 0.0% joint accuracy in Task 2 is worse than random guessing and indicates a consistent failure mode, not evaluation noise. We also added a dedicated **task solvability** discussion (Section 4 / Appendix), showing that: (i) Task 1 is deterministic (spherical truncation of a periodic lattice), (ii) Task 2 is theoretically solvable because local neighborhoods retain crystallographic invariants (coordination shells, bond-angle statistics, stoichiometry), and (iii) Task 3 has a clear lower bound, with oracle-style baselines (e.g., nearest-neighbor lattice filling) achieving significantly lower error than current learned models. This clarifies that the tasks are challenging by design but solvable in principle, and that the observed failures reflect current architectural limitations rather than ill-posed objectives.

**Rotations, notation, and implementation details.** We clarified why rotations are an intentional part of the benchmark rather than a nuisance to be aligned away: non-overlapping SO(3) grids for train/ID/OOD are used specifically to test orientation generalization, and Kabsch/ICP alignment would artificially remove that difficulty. Notation for radius, rotations, and orientation indices (R, (\mathcal{R}), (b)) has been cleaned up, and we added a short symbol table. We also expanded the implementation details (Appendix) to explain how periodic models like DiffCSP and FlowLLM are adapted to finite nanoparticles, how nanoparticle→lattice inference and masked reconstruction are implemented in practice, and how SchNet-type encoders scale to our largest systems (with neighbor cutoffs and batching). FlowLLM training budgets, parameter sizes, and losses are now explicitly documented and kept compute-comparable to other baselines.

**Related work, symmetry-aware models, and nanostructure ML.** We expanded the Related Work section to include recent symmetry-preserving crystal generators and nanostructure-focused ML papers, including Space Group Constrained Crystal Generation, SymmCD, WyckoffDiff, and diffraction-based structure-solution work \cite{guo2025ab}. We explain why periodic, Wyckoff/space-group–based generators are not directly applicable to finite, non-periodic nanoclusters without substantial re-engineering, and we explicitly list benchmarking these adapted models as future work. We also clarified how StructEval complements existing property-focused benchmarks (e.g., MatBench, OC20/OC22, CrysMTM) by deliberately omitting energetic labels to isolate structural and symmetry reasoning.

**Scope, realism, and future extensions.** We justified the choice of ten materials as a balance between chemical diversity and computational tractability (~200k structures up to 11,298 atoms each) and added a note that the generation pipeline is general and future releases will extend to more chemistries and phases. We clarified that nanoparticles are carved from experimentally validated bulk crystals using a deterministic spherical truncation pipeline, preserving realistic local coordination, and we added discussion on how StructEval can be extended to XRD-style tasks and joint real-space/reciprocal-space modeling in future work.

Overall, we believe these revisions significantly improve clarity, strengthen the ML-centric framing, and directly address concerns about evaluation validity and task solvability.

Sincerely,
Authors.

---

### Note · Authors · 2026-01-27

I have read and agree with the venue's withdrawal policy on behalf of myself and my co-authors.

---

### Meta-Review · Area_Chair_Uk7D · 2025-12-31

**Summary:**

This submission introduces StructEval, a benchmark intended to evaluate generative modeling across periodic unit cells and finite nanoparticles, with explicit in-distribution and out-of-distribution splits over orientation and radius. The authors benchmark several existing methods and report substantial performance degradation under the OOD setting.

The motivation is clear, and a unified benchmark across periodic and finite regimes could be valuable. However, questions around the interpretability and robustness of the benchmark results remain, particularly whether the observed failures reflect true generalization gaps rather than evaluation artifacts or model-task mismatch. The benchmark’s claims of broad relevance may not yet be sufficiently supported by the current evidence which focuses on a subset of generation tasks with particular conditions.

**Reviewer Concerns:**

Reviewer 7n8K:
The rebuttal addressed issues around formatting, notation, clarification of rotations, task definitions, and baseline adaptation. However, the reviewer’s core concern might remain, such as the repeated metrics and extreme failures, e.g., 0% accuracy, which are explained as “mean collapse,” but no evidence was provided, and no evidence was provided to rule out evaluation artifacts or model-task mismatch. It remains unclear whether the benchmark reliably measures the intended generalization gaps.

Reviewer fmVf:
The authors expanded related work, added a solvability discussion, and clarified training details. The major concerns about the generality and completeness of the benchmark, particularly the limited coverage of strong symmetry-aware approaches remain. The concern around the reliance on mostly conceptual solvability arguments rather than empirical baselines might also remain.

Reviewer seei (borderline):
Scope-related concerns such as the number of materials and the lack of XRD or property prediction tasks were reasonably justified, but the benchmark’s claims of broad relevance may not yet be sufficiently supported by the current evidence.

**Reviewer Scores:**

Reviewer 7n8K: likely remains reject. The rebuttal may not eliminate their skepticism about the results and evaluation validity.

Reviewer fmVf: likely remains the same. The rebuttal (and the proposed benchmark) didn't provide additional experiments the reviewer requested such as generating infinite periodic crystals with space groups and Wyckoff positions.

Reviewer seei: likely remains the same or increases to weak accept.

---

### Decision · Program_Chairs · 2026-01-26

Reject